# The impacts of climate change on hydrological processes of Gilgel Gibe catchment, southwest Ethiopia

**Zewde Alemayehu Tilahun** *, **Yechale Kebede Bizuneh, Abren Gelaw Mekonnen**

Dep't of Geography & Env'tal Studies, Arba-Minch University, Arba Minch, Ethiopia

* zedoalex8@gmail.com

**Data Availability Statement:** The data that support the findings of this study are available from various sources. The Gilgel Gibe catchment digital elevation model (DEM) was extracted from the Shuttle Radar Topography Mission (SRTM) 30 meter resolution

## Abstract

Climate change is a significant driver of water resource availability, affecting the magnitude of surface runoff, aquifer recharge, and river flows. This study investigated the impact of climate change on hydrological processes within the Gilgel Gibe catchment and aimed to determine the level of exposure of water resources to these changes, which is essential for future adaptability planning. To achieve this objective, an ensemble mean of six regional climate models (RCMs) from the coordinated regional climate downscaling experiment (COR-DEX)-Africa was used to simulate future climatic scenarios. The RCMs outputs were then bias corrected using distribution mapping to match observed precipitation and temperature. The Soil and Water Assessment Tool (SWAT) model was used to assess the hydrological impacts of climate change on the catchment. The results indicated that the ensemble mean of the six RCMs projects a decline in precipitation and an increase in temperature under both the RCP4.5 and RCP8.5 representative concentration pathways. Moreover, the increases in both maximum and minimum temperatures are higher for higher emission scenarios, indicating that RCP8.5 is warmer than RCP4.5. The projected climate change shows a decrease in surface runoff, groundwater, and water yield, resulting in an overall decline of annual flow. This decline is mainly due to the reduction in seasonal flows driven by climate change scenarios. The changes in precipitation range from -11.2% to -14.3% under RCP4.5 and from -9.2% to -10.0% under RCP8.5, while the changes in temperature range from 1.7˚C to 2.5˚C under RCP4.5 and from 1.8˚C to 3.6˚C under RCP8.5. These changes could lead to reduced water availability for crop production, which could be a chronic issue for subsistence agriculture. Additionally, the reduction of surface water and groundwater could further exacerbate water stress in the downstream areas, affecting the availability of water resources in the catchment. Furthermore, the increasing demands for water, driven by population growth and socioeconomic progress, along with the variability in temperature and evaporation demands, will amplify prolonged water scarcity. Therefore, robust climate-resilient water management policies are indispensable to manage these risks. In conclusion, this study highlights the importance of considering the impact of climate change on hydrological processes and the need for proactive adaptation measures to mitigate the impacts of climate change on water resources.

data from USGS website (https://earthexplorer.usgs.gov). Daily weather data (1991–2021) were used from nine precipitation and temperature recording stations located within and around the study Catchment. The daily data were collected from the National Meteorological Agency (NMA) of Ethiopia. Daily discharge data of eight gauging stations for the period 1991 to 2021were collected Ministry of Water and Energy of Ethiopia (MoWE). Software used ArcGIS, AcrSWAT, ENVI, GIS.

**Funding:** The authors received no specific funding for this work.

**Competing interests:** The authors have declared that no competing interests exist.

# 1. Introduction

The world's environment is a complex system of interrelated components that operate at varying spatial and temporal scales. The interactions between these components, which include both natural and human factors, are highly intricate and can have far-reaching impacts on the health and wellbeing of the planet [1].

The concentrations of greenhouse gases (GHGs) in the atmosphere are widely believed to be increasing due to human activities, particularly the combustion of fossil fuels and land use and land cover changes [2]. This change in atmospheric gas composition has resulted in a warming effect on the atmosphere, which is causing climate change [3]. The projected changes in precipitation and temperature are consistent with the global and regional trends reported by IPCC [4] in its Sixth Assessment Report [4]. According to [4], human influence has warmed the climate system since pre-industrial times at an unprecedented rate across all regions [4]. This warming has led to changes in precipitation patterns and intensity, as well as increased frequency and intensity of some extreme events such as heat waves, droughts, and heavy rainfall [4]. Extreme hydrological events such as floods and droughts are significant contributors to natural disasters in many regions of the world [5]. According to the Clausius-Clapeyron theory, the frequency of extreme events is expected to rise as a result of climate change [6]. Changes in precipitation and hydrological processes are also expected to have significant impacts on ecosystems, agricultural production, and water resources [7].

Numerous studies have been conducted to investigate the potential impacts of climate change on hydrological processes, with significant variations observed depending on the utilized climate model, emission scenarios, and observed spatial variability [8, 9]. Despite being one of the most vulnerable regions to climate change and climate variability, there have been limited studies examining the effects of climate change on hydrological processes in Sub Saharan Africa [10]. [11] found a significant increase in flow volume in the Mara River basin in Kenya/Tanzania during the years 2046 to 2065 and 2081 to 2100, while [12] projected an annual rise in flow volume in the Gilgel Abay River in Ethiopia from 2070 to 2100. Other studies, such as [13, 14], have projected both increases and declines in streamflow for the Dinder River in Sudan and the Jemma sub-basin of the upper Blue Nile Basin, respectively. These fluctuations are caused by erratic and unpredictable changes in climate factors, leading to seasonal and yearly flow changes and declines in specific catchments [15]. To effectively manage water resources and the environment in the face of these changes, there is a need for scientific research to understand the linkages between climate change and its effects on hydrological processes. Moreover, new insights on water and land conditions, along with updated management options, can facilitate proactive approaches to maintain water resources and land health while reducing degradation risk.

The escalating global population has heightened the need for increased water extraction to supply to the agricultural, urban, industrial, and environmental sectors [16]. Consequently, it is imperative to comprehend the hydrological processes associated with land surface under current and future climate variability for effective adaptation to climate change and its impacts. Such foundational knowledge will provide a basis for the development of sustainable land and water management practices.

A comprehensive understanding of water resources development, informed by projected climate changes, is becoming increasingly necessary for sustainable water resources management [17]. In order to design new water resource management strategies that are resilient to these changes, it is imperative to address the uncertainties associated with a changing climate, in addition to the inherent natural variability that is already reflected in conventional water planning systems [18].

Research into the effects of climate change in Ethiopia has revealed a troubling trend of rising pressures on the availability of water resources, both seasonally and annually [19, 20]. These changes have significant implications for the country's agricultural sector, which is heavily dependent on water resources for irrigation and cultivation. Specifically, studies conducted in Ethiopia's Rift Valley have shown a decrease in precipitation and an increase in temperature, exacerbating the already precarious situation of water scarcity in the region [21]. These changes have a cascading effect on the environment, impacting soil quality, biodiversity, and ecosystem health. The situation is further compounded by the growing demand for water resources in Ethiopia due to population growth and socio-economic factors. As such, there is an urgent need for coordinated efforts aimed at mitigating the effects of climate change on water resources in the country. Understanding the impacts of climate change on water resources is essential for developing sustainable water management strategies that can support the needs of both current and future generations. It is imperative that we continue to prioritize research into this critical issue and work collaboratively to develop effective solutions that can safeguard the future of Ethiopia's water resources.

Research on the effects of climate change on watershed hydrology is increasingly important for understanding the implications and vulnerabilities of climate change, as well as developing sustainable water resource management strategies [22–25]. However, there is a lack of understanding of the implications of climate change on hydrology at the catchment level. The Gilgel Gibe catchment is of particular concern due to its intensive use and vulnerability to environmental degradation [26–28]. The Gibe River is a vital resource for generating hydropower and providing freshwater to small towns along its course [29]. Thus, further research is needed to assess the impact of climate change on hydrological processes within the catchment and inform sustainable water resource management practices.

The Gilge Gibe catchment, located in the Omo Gibe basin, is facing a number of challenges due to rapid population growth, expansion of cultivation lands, and urban expansion within the catchment [30]. These factors have led to unprecedented impacts on the catchment, exacerbating the high demand for water resources due to socio-economic progress and increasing demand for cultivation [31]. Despite this, there has been little investigation into the impact of climate change on hydrological processes at the catchment level, particularly with regards to the Gilgel Gibe catchment, which is a major tributary to the Gibe river. To address this knowledge gap, regional and local-scale studies are required to evaluate the effects of climate change on hydrological processes in the specific area. This research aims to investigate the impact of climate change on hydrological processes in the Gilgel Gibe catchment using the ensemble mean of six regional climate models (RCMs) from the Coordinated Regional Climate Downscaling Experiment (CORDEX)-Africa.

The findings of this study will provide valuable insights into the vulnerability of the Gilgel Gibe catchment to the effects of climate change. By assessing both current and future impacts, the research will offer a comprehensive understanding of the hydrological consequences of climate change in this region. Ultimately, this research has important implications for water resource management and environmental policy in Ethiopia, providing policymakers and stakeholders with the necessary information to develop strategies and interventions aimed at mitigating the impact of climate change on water resources. The objective of this research is to comprehensively investigate the impact of climate change on the hydrology of the Gilgel Gibe catchment. The study is designed to achieve specific aims, including: 1. Assessing and modeling the climate change that has occurred in the Gilgel Gibe catchment, and determining its impacts on hydrological processes. 2. Evaluating the potential impact of future climate change on the hydrology of the catchment, using established models and methodologies. 3. Exploring the sensitivity of water resources in the Gilgel Gibe catchment to changing climate conditions.

## 2. Study area and research methods

### 2.1 Description of the study area

**2.1.1 Location.** The study area of this research is Gilgel Gibe Catchment, located in the upstream of the Omo Gibe basin in Jimma zone of Oromia National Regional State, located approximately 260 km southwest of Addis Ababa. The geographical coordinates of the catchment are approximately $7^0 36'0''$ - $8^0 0'0''$ N latitude and $36^0 36'0''$ - $37^0 34'0''$E longitude, as shown in Fig 1 (ArcGIS 10.5). The catchment encompasses an area of approximately 514,103.4 ha, with elevations ranging from 1,096 to 3,341 m above sea level. The catchment spans across several districts, including Sekoru, Tiro Afeta, Kersa, Seka Chekorsa, Omo Nada, Dedo, Shebe Sembo, and Chora in Jimma zone, as well as Jimma special town. The Gilgel Gibe River is the main river of the catchment and is intersected by the Gilgel Gibe hydroelectric dam.

**2.1.2 Soil.** The topography of the Gilgel Gibe catchment ranges from flatlands in the valley bottom to steep slopes in the surrounding mountain ranges. The catchment is dominated by various soil types according to FAO/UNESCO [32], including Chromic Vertisols, Chromic Vertisols/Eutric Fluvisols, Dystric Nitoso, Chromic Vertisols/Eutric Nitosols, Dystric Nitosols, Dystric Nitosols/Orthic Acrisols, Eutric Nitosols/Chromic Vertisols, Lithosols/Eutric Cambisols, Orthic Acrisols/Dystric Nitosols, and Pellic Vertisols (Fig 2).

**2.1.3 Climate.** Despite being located in the tropical zone, Ethiopia exhibits a diverse range of climates due to several factors. Altitude, rift valley, and the Inter-Tropical Convergence

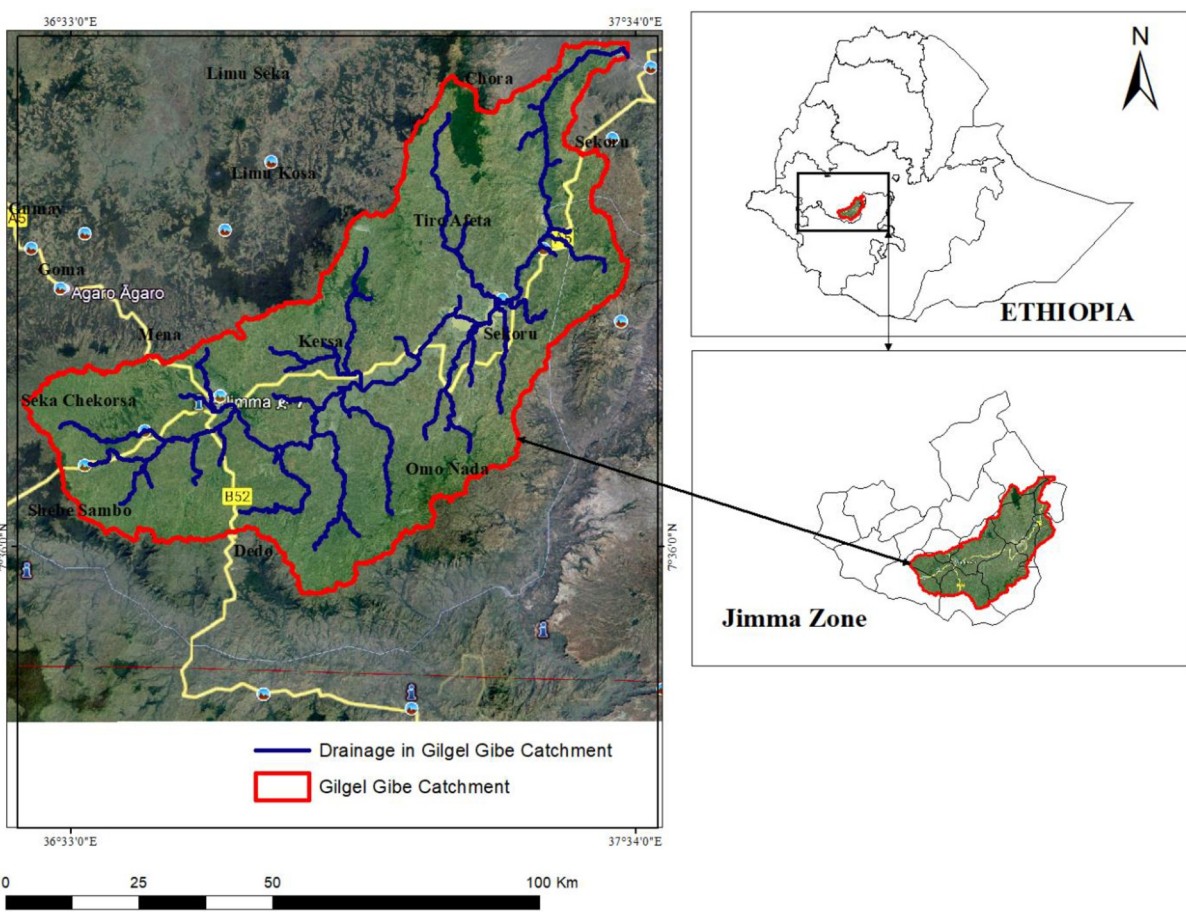

**Fig 1. Location and elevation map of study area.**

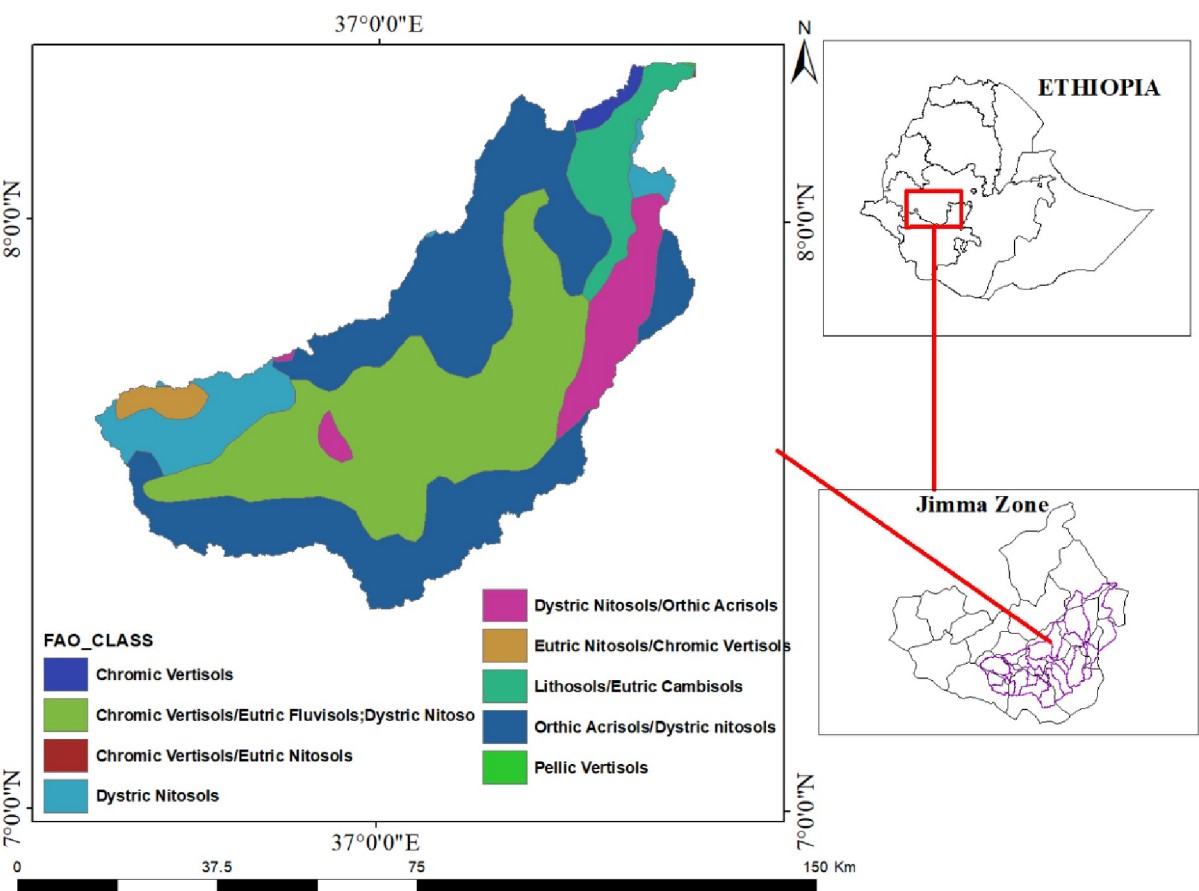

**Fig 2. The dominant soils of Gilgel Gibe catchment.**

Zone (ITCZ) all influence temperature and precipitation patterns across the country. Average temperatures can range from below 10°C to above 40°C, with annual rainfall varying from 100mm to 2,800mm (Fig 3) depending on the location. Additionally, rainfall patterns differ across Ethiopia, with some areas experiencing two rainy seasons (Belg from March to May and Kiremt from June to August) and others just one [33]. On average, minimum and maximum temperatures are 11.5°C and 27.5°C, respectively (Fig 4), with an average precipitation of 1,521 mm. The lower reaches of the basin receive around 1,300 mm of precipitation, while the upper reaches receive up to 2,000 mm [33].

## 2.2 Research methods

**2.2.1 General circulation model (GCM) data.** Daily rainfall and temperature time series were obtained from runs with six different new generations of General Circulation Models (GCMs) from the Coupled Model Intercomparison Project Phase 5 (CMIP5) database of the Intergovernmental Panel on Climate Change (IPCC) Data Distribution Center (http://cmip-pcmdi.llnl.gov/cmip5/). The GCMs considered for this study are shown in Table 1, and the data covers the period 1991–2021 for control simulations. For future simulations, two periods were considered: the intermediate period (2041–2070) and far future period (2071–2099). The future scenarios considered for this study are based on the IPCC Fifth Assessment Report (AR5) [34], which includes four Representative Concentration Pathways (RCPs): RCP 8.5, RCP 6, RCP 4.5, and RCP 2.6. This study implemented a moderate climate change scenario

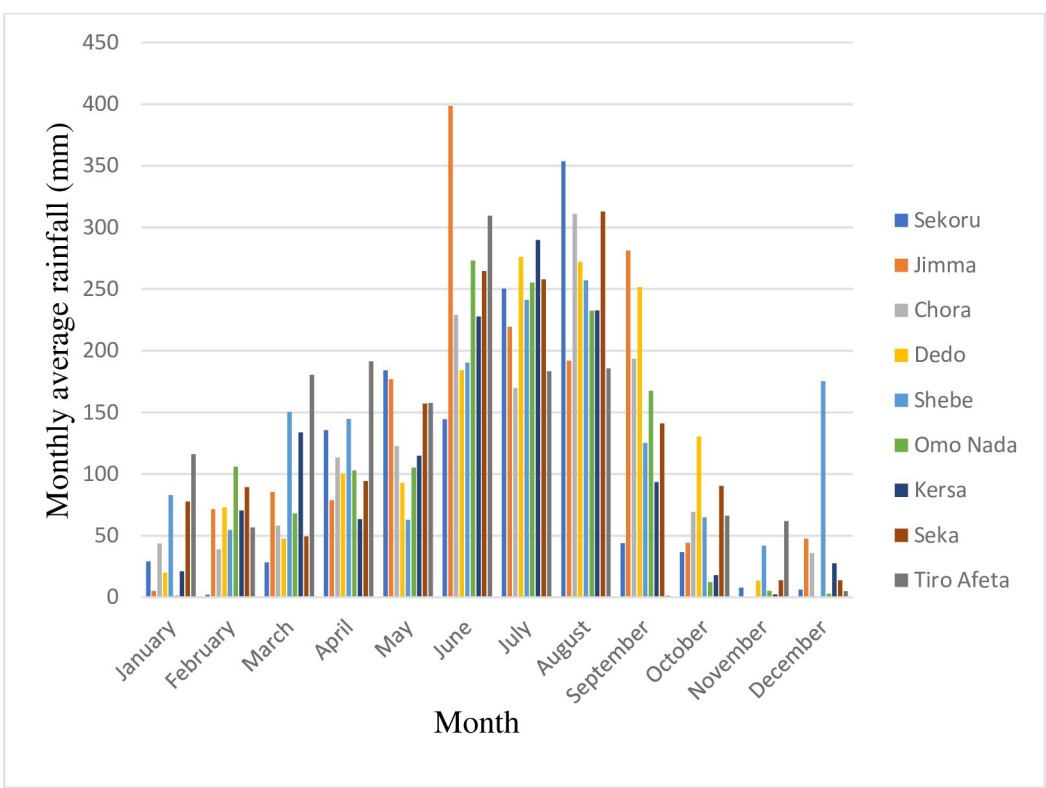

**Fig 3. Average monthly rainfall of the selected stations.**

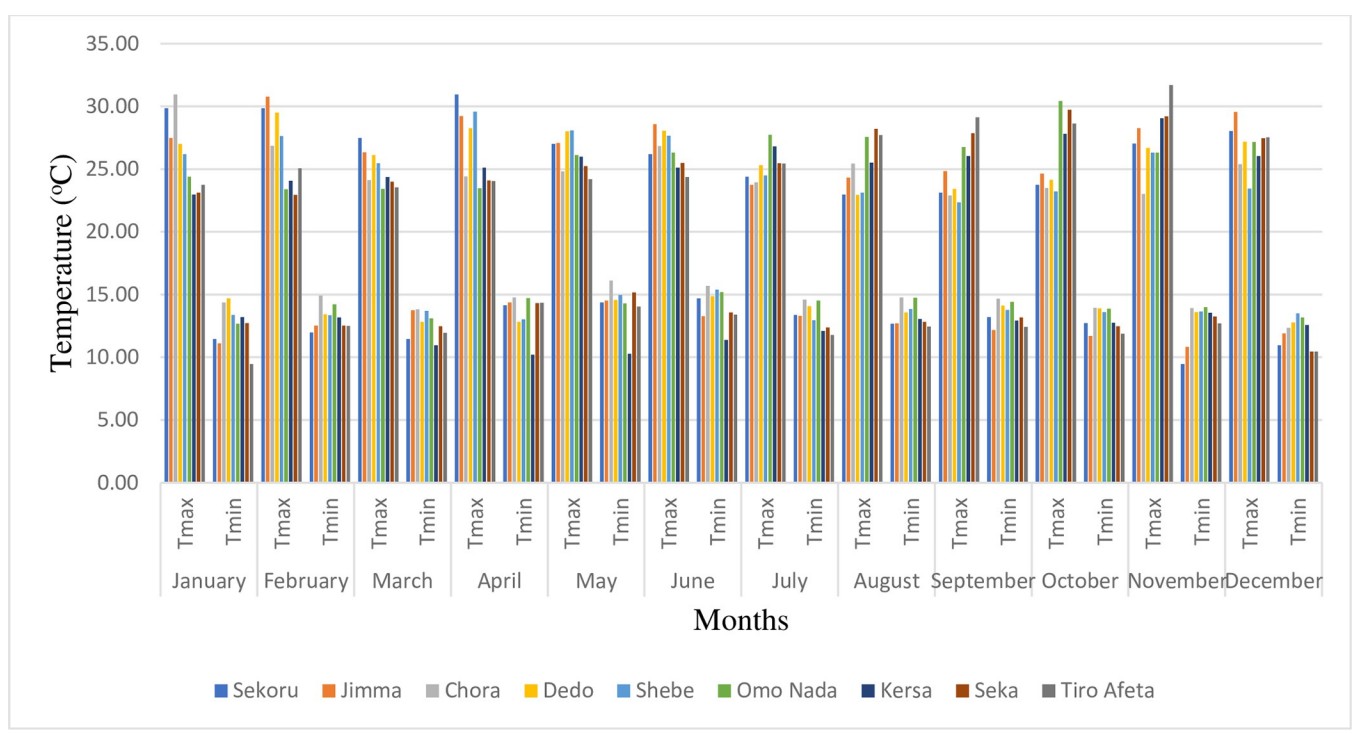

**Fig 4. Monthly maximum and minimum temperature of the selected stations.**

**Table 1. Description of the climate models (RCMS) from which data is obtained for this study.**

| No | RCMs | Model Center |
|----|------|--------------|
| 1 | CCLM4-8 | Climate Limited Area modelling Community (CLMcom), USA |
| 2 | HIRHAM5 | Denmarks Meteorologiske institute (DMI), Denmark |
| 3 | RACMO22T | Koninklijk Nederlands Meteorologisch Instituut (KNMI), Netherlands |
| 4 | RCA4 | Sveriges Meteorologiska och Hydrologiska institute (SMHI), Sweden |
| 5 | CRCM5 | Universite du Quebec a Montreal (UQAM), Canada |
| 6 | REMO2009 | Max Planck Institute for Meteorology-Climate Service centre (MPI-CSC), Germany |
|  | Ensemble | Ensemble mean of all RCMs |

(RCP 4.5) and an extreme scenario (RCP 8.5) to detect the possible impact of climate change. A summary of RCMs projections is presented in Table 1.

**2.2.3 Climate change scenarios.** The performance of the Regional Climate Models (RCMs) was evaluated using the root mean square error (RMSE) and the Nash-Sutcliffe efficiency (NSE), which measure the average error and the model's ability to reproduce the observed variability and distribution of the values, respectively [35, 36]. The evaluation followed the guidelines of the Coordinated Regional Climate Downscaling Experiment (CORDEX) and showed that the RCMs performed reasonably well, with the ensemble mean of the models performing better than the individual models. The RMSE and NSE for the ensemble mean were all within acceptable ranges, indicating that the models were able to capture the main features of the observed climate. These results are consistent with those of other studies that have used RCMs for climate change impact assessment. Following an evaluation of Regional Climate Models (RCMs) as part of the Coordinated Regional Climate Downscaling Experiment (CORDEX) Africa, six RCMs that demonstrated better performance were selected. This is consistent with a similar study by [35] in the central Rift valley basin where the ensemble mean of five RCMs showed better performance than individual RCMs. The use of multi-RCMs for climate modelling helps to reduce uncertainties compared to the use of a single RCM. However, despite the reasonable performance of the RCMs, including the ensemble mean, all RCMs displayed noticeable biases. Therefore, it is necessary to correct these biases before conducting a climate change impact study.

In this study, six Regional Climate Models (RCMs) were used, namely CCLM4-8, HIRHAM5, RACMO22T, RCA4, CRCM5, and REMO2009. The future scenarios were evaluated for the intermediate (2041–2070) and far future (2071–2099) periods, while the period from 1991 to 2021 was used as the historical baseline to evaluate climate changes. Other weather variables, such as solar radiation, relative humidity, and wind speed, were considered in the future scenarios without modification, as changes in these variables may not significantly impact the modeling of climate change scenarios on local hydrology.

**2.2.4 Bias correction.** The climate model data for hydrological modelling (CMhyd) [37], obtained from https://swat.tamu.edu/software/, was used to process the precipitation and temperature bias correction. [38] have provided a full review of the bias correction techniques. According to [38] all the bias correction techniques have improved the simulation of precipitation and temperature. However, there are differences between the correction methods in the daily precipitation series, standard deviations, and percentiles. In our analysis, we evaluated multiple bias correction methods, including the distribution mapping method, delta change, scaling factor, quantile mapping and empirical quantile mapping. We compared the performance of these methods for both temperature and precipitation corrections based on mean absolute error ranking. After carefully considering these various bias correction methods, we concluded that the distribution mapping method provided the most accurate results.

The distribution mapping uses a transfer function to adjust the cumulative distribution of estimated data to the cumulative distribution of rain gauges [39, 40]. applied seven precipitation bias correction methods and five methods for temperature. The results showed that distribution mapping reproduces precipitation and temperature very well. A similar study by [41] compared five bias correction methods, using CMhyd, and found distribution mapping performed best for climate change impact study on streamflow dynamics of two rivers in Northern lake Erie basin, Canada. The study by [42] used CMhyd for extraction of CORDEX-NetCDF, and bias correction of minimum and maximum temperature to predict climate change-induced temperature changes in Finchaa catchment. The study compared distribution mapping with other methods such as the delta change, scaling factor and quantile mapping, and demonstrated that distribution mapping was more effective in improving the accuracy of predictions by adjusting the distribution of temperature and precipitation values. This was achieved through the utilization of historical data to calibrate the simulation or by employing statistical methods to modify the distribution of values. Owing to all the above findings, distribution mapping served as a basis for the precipitation and temperature corrections before using CORDEX-RCMs outputs for a climate impact study.

## 2.3 Hydrological modelling

The hydrological modelling component of this study used Soil and Water Assessment Tool (SWAT), Soil and Water Assessment Tool (SWAT) is a physically-based semi-distributed model that operates on a continues time scale [43]. SWAT model operates on a daily time step and predicts the impact of land management in large complex watersheds with varying soils, land use and management conditions. Major model components include DEM, weather, hydrology, soil properties and land management [44]. In SWAT, a watershed is divided into multiple sub-watersheds, which are then further subdivided into Hydrologic Response Units (HRUs) that comprise homogeneous land use, slope and soil characteristics. The SWAT interface used an ArcSWAT, a SWAT application within ArcGIS. The hydrologic cycle as simulated by SWAT is based on the water balance equation as developed by [44].

$$S_{wt} = S_{wo} + \sum_{i=1}^{t} (R_{day} - Q_{surf} - E_a - W_{seep} - Q_{gw}) \qquad (1)$$

Where $S_{wt}$ is the final soil water content (mm), $S_{wo}$ is the initial soil water content (mm), t is time (days), $R_{day}$ is the amount of precipitation on day i (mm), $Q_{surf}$ is the surface runoff on day i (mm), $E_a$ is the amount of evapotranspiration on day i (mm), $W_{seep}$ is the amount of percolation and bypass flow exiting the soil profile bottom on day i (mm), and $Q_{gw}$ is the amount of return flow on day i (mm).

The ArcSWAT evolved from ArcSWAT2012 is an ArcGIS extension developed for an earlier version of SWAT. It provides a graphical user interface that allows for GIS data to be easily formatted for use in SWAT model simulations. In the SWAT model, the simulation of the hydrological process begins with watershed delineation and generating streamflow networks. Using a 30 m resolution Digital Elevation Model (DEM), the Gilgel Gibe catchment was divided into 23 sub-watersheds. To further characterize the catchment, a hydrologic response unit (HRU) was established, taking into account different attributes. Specifically, classes representing 10% land use, 20% soil, and 10% slope were assigned. As a result, a total of 428 hydrologic response units (HRUs) were established based on these attribute percentages. The weather data inputs from nine stations (Sekoru, Jimma, Chora, Dedo, Shebe, Omo Nada, Kersa, Seka, and Tiro Afeta) was utilized. Among these stations, Sekoru and Jimma were chosen as weather generators using an ArcSWAT weather generator (WGEN) due to their comprehensive coverage of all climate variables required for the SWAT model setup.

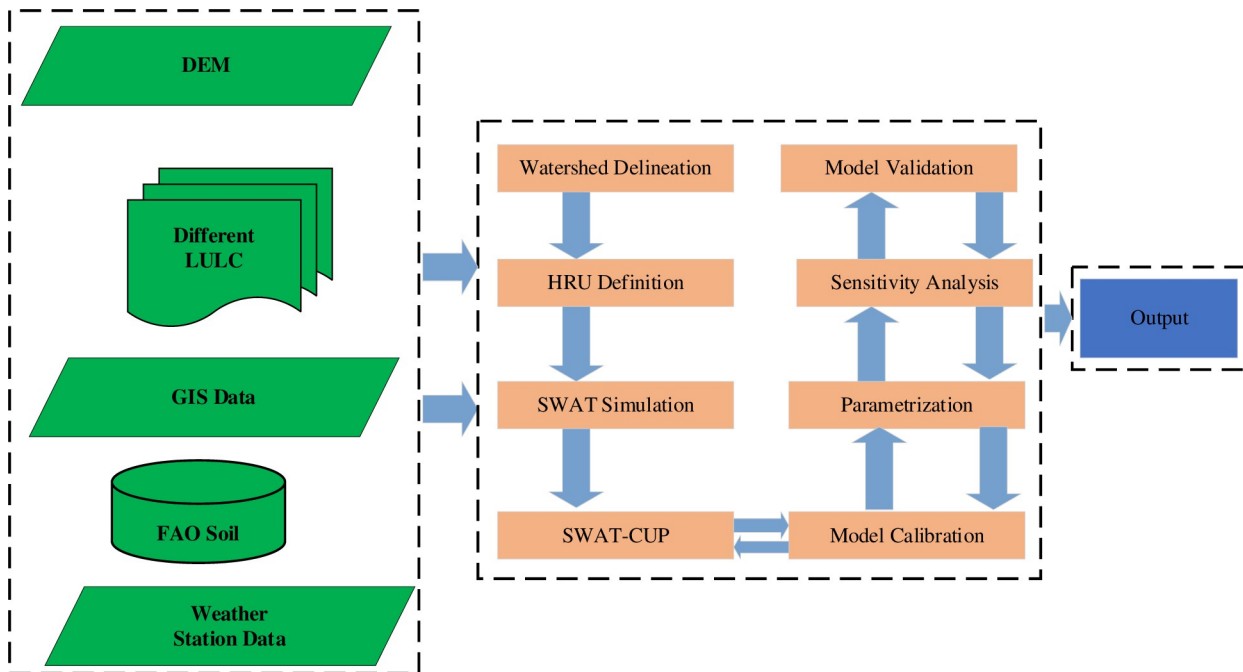

**Fig 5. Flowchart of ArcSWAT processing steps for the Gilgel Gibe catchment.**

Before using the output of the SWAT simulation for analysis, the performance of the model was evaluated for the catchment. It is important to understand the conceptual framework of each step, as well as what data are used and how they are integrated into ArcSWAT. Fig 5 shows the flowchart of modelling using ArcSWAT.

**2.3.1 Model inputs.** *2.3.1.1 Watershed.* The Gilgel Gibe catchment digital elevation model (DEM) was extracted from the Shuttle Radar Topography Mission (SRTM) 30 meter resolution data from USGS website (https://earthexplorer.usgs.gov) Watershed boundary, sub-watersheds (SWs), hydrological response units (HRUs) and slope layers were defined using these DEM data. Delineation of the watershed has been made by considering the gauging station as an outlet. The watershed boundary was further divided into 23 sub-watersheds (SWs), which were then reapportioned into 428 HRUs to allow spatially detailed simulation by reflecting differences in various hydrological conditions for different LULC, soils and slope configurations (Fig 6).

*2.3.1.2 Land use and land cover.* Landsat Thematic Mapper of 1991, Enhanced Thematic Mapper Plus of 2006, and Operational Land Imager or Thermal Infrared Sensor of 2021 were used for LULC classifications. The mosaic of Worldwide Reference System (WRS) Path 169, Row 54, Path 169, Row 55 and Path 170, Row 55 were used to extract the LULCs layers. Supervised image classification method recommended in [45, 46] were employed to interpret and classify the images. The 2021 satellite image was utilized for the purpose of performing a land use and land cover (LULC) classification, specifically for the SWAT model.

*2.3.1.3 Soil.* The SWAT model requires soil data to determine the hydrological parameters of each soil category within each SW and HRU. The major soil physicochemical characteristics such as soil depth, soil hydrological group, soil texture, bulk density, available water capacity, saturated hydraulic conductivity, organic carbon content, soil albedo, rock fragments and soil erodibility factors were considered in the analysis. The input soil layers in the ArcSWAT model were extracted from the Food and Agriculture Organization of the United Nations (FAO) [32, 47] and Harmonized World Soil Database (HWSD) [48].

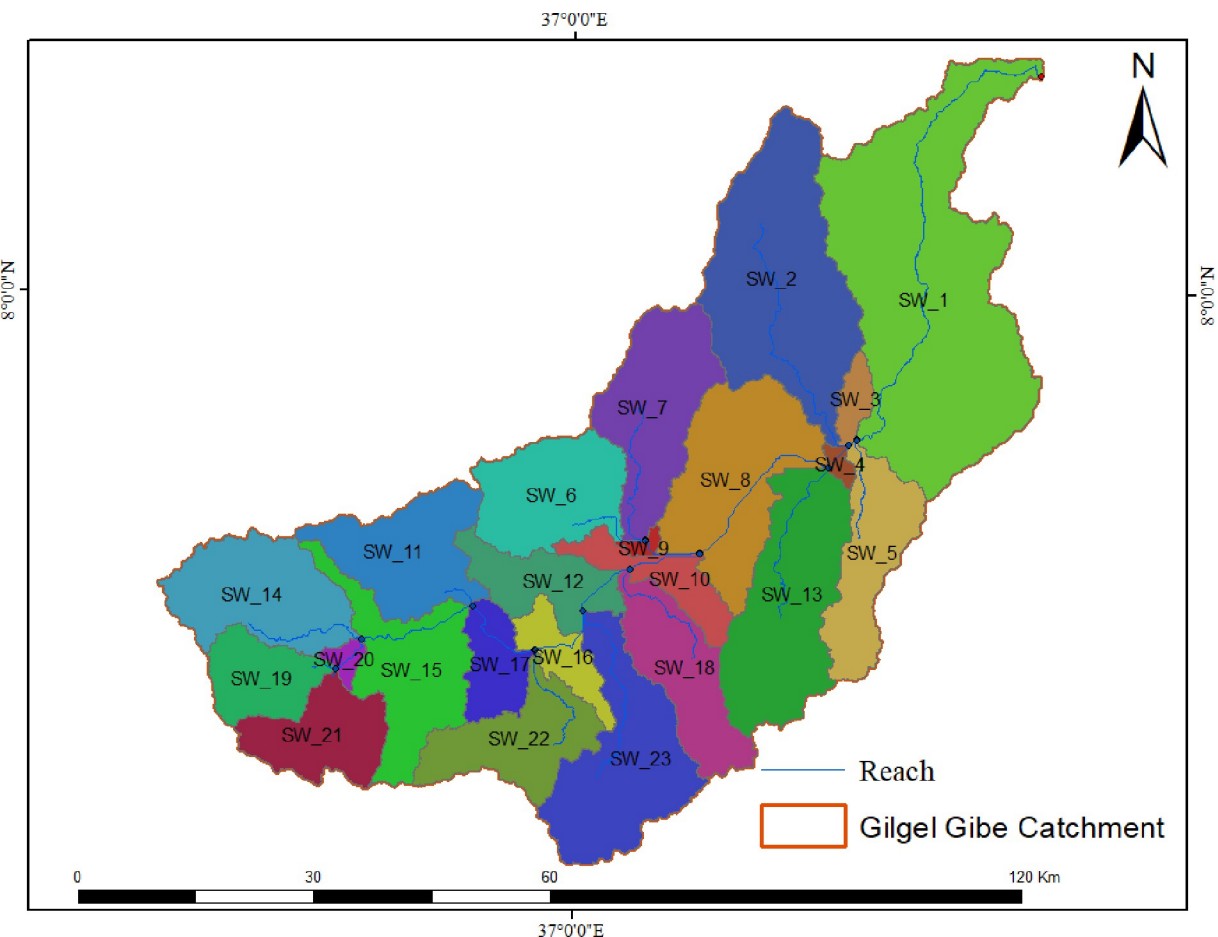

**Fig 6. Sub-watersheds of the Gilgel Gibe catchment.**

*2.3.1.4 Weather.* Daily weather data (1991–2021) were used from nine precipitation and temperature recording stations located within and around the study Catchment (Sekoru, Jimma, Chora, Dedo, Shebe, Omo Nada, Kersa, Seka and Tiro Afeta) (Fig 7). The daily data were collected from the National Meteorological Agency (NMA) of Ethiopia. However, some missing weather elements such as precipitation, maximum and minimum temperature, wind speed, solar radiation and relative humidity data were generated using an ArcSWAT weather generator (WGEN). Moreover, weather data from 1991 and 1992 were used as the model initialization phase (warm up) and not included in the final analysis.

*2.3.1.5 Discharge.* In Gilgel Gibe catchment drainage system (Fig 8); the Gilgel Gibe catchment river discharge gauging stations, Daily discharge data of Gibe near Seka, Bulbul near Serbo, Gilgel Gibe at Abelti, Gilgel Gibe near Asendabo, Gojeb near Shebe, Bidru Awana near Sekoru, Kito near Jimma and Awetu at Jimma gauging stations (Fig 9) for the period 1991 to 2021 were collected Ministry of Water and Energy of Ethiopia (MoWE). These data were used for model calibration and validation. Model calibration was performed for the period from 1993 to 2011, whereas validation was performed for the period from 2012 to 2021.

*2.3.1.6 Sensitivity analysis, calibration and validation.* Owing to a large number of flow parameters in SWAT, identifying the most sensitive parameters is necessary to improve the calibration of the hydrological model. Model calibration and validation was undertaken automatically using SWAT-CUP (Calibration and Uncertainty Programs). The most sensitive

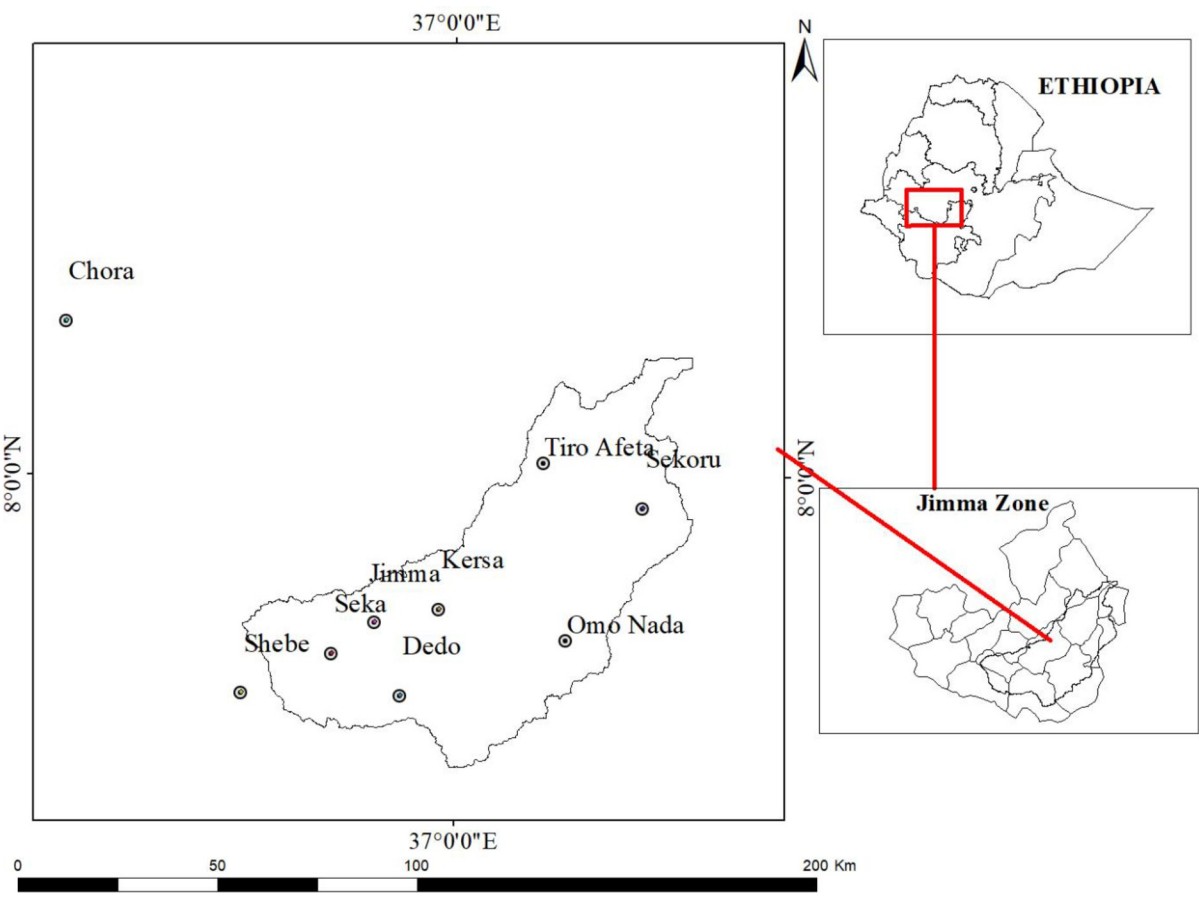

**Fig 7. Precipitation and temperature recording stations.**

parameters that have a strong influence on the flow process were identified through the sensitivity analysis. The Sequential Uncertainty Fitting (SUFI-2) embedded in the SWAT-CUP (Calibration and Uncertainty Program) was used to achieve the sensitivity analysis, calibration and validation [49].

Calibration of the hydrological model is the process of estimating model parameters by comparing the model prediction with the observed data for the same condition [49, 50]. Calibrations are very critical for parameters that were not measured and are intrinsically heterogeneous and uncertain, as it serves to optimize the unknown model parameters. For model calibrations, rules of parameter regionalization given by [51] was used. Validation was used to test the calibrated model without further parameter adjustments with an independent dataset. Observed streamflow of 1991–2021 was split into a warm-up (1991–1992), calibration period (1993–2011) and validation period (2012–2021) (Fig 10).

The fitness of the model simulation with the observed streamflow was expressed by statistics like coefficients of determination ($R^2$), Nash-Sutcliffe efficiency (NSE), percent bias (PBIAS) and the ratio of the root-mean-square error to the standard deviation of measured data (RSR). The model performance ratings were based on the statistics recommended by [50, 52] $R^2$ varies between 0 and 1, where higher value shows less error. NSE ranges from negative infinity to 1 (best). PBIAS close to 0 shows best the simulation, a negative value indicates overestimation and a positive value indicates under simulation of the model. RSR varies from zero to a large positive number; the lower RSR shows a better simulation of the model.

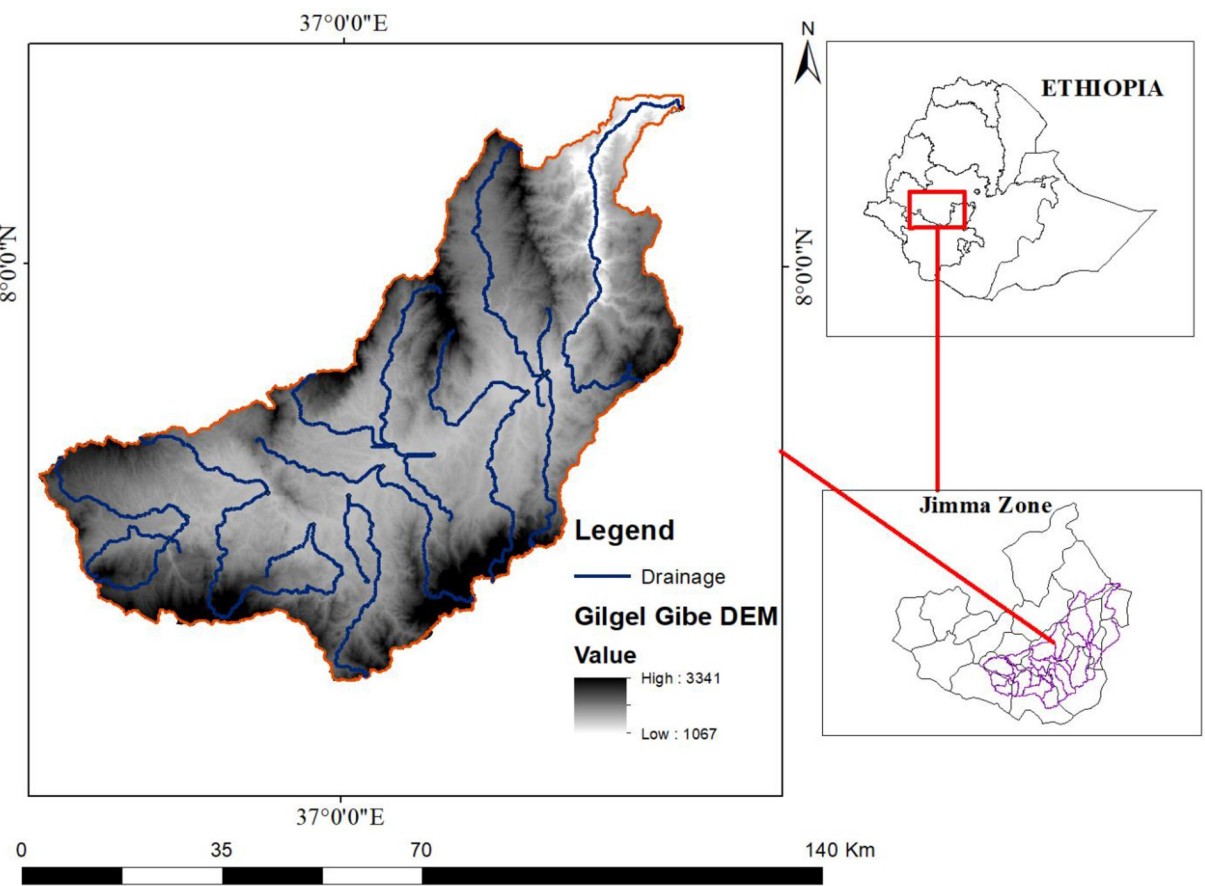

**Fig 8. Drainage in Gilgel Gibe catchment.**

Furthermore, for the sensitivity of each parameter t-test and p-values was considered, using a t-test (larger in absolute values are more sensitive), and p-values were used to determine the significance of the sensitivity (a value close to zero has more significance) [49]. Statistical values of $R^2$, NSE, PBIAS and RSR were computed using Eqs 2, 3, 4 and 5 respectively:

$$R^2 = \left\{ \frac{\sum_{i=1}^{n}(O_i - \bar{O})(P_i - \bar{P})}{[\sum_{i=1}^{n}(O_i - \bar{O})^2]^{0.5}[(\sum_{i=1}^{n}(P_i - \bar{P})^2]^{0.5}} \right\}^2 \tag{2}$$

$$NSE = 1 - \left[ \frac{\sum_{i=1}^{n}(O_i - P_i)^2}{\sum_{i=1}^{n}(O_i - \bar{O})^2} \right] \tag{3}$$

$$PBIAS = \left[ \frac{\sum_{i=1}^{n}(O_i - P_i)}{\sum_{i=1}^{n}(O_i)} *100 \right] \tag{4}$$

$$RSR = \frac{RMSE}{STDEV_{obs}} = \left[ \frac{\sqrt{\sum_{i=1}^{n}(O_i - P_i)^2}}{\sqrt{\sum_{i=1}^{n}(O_i - \bar{O})^2}} \right] \tag{5}$$

Where $O_i$ is the $i^{th}$ observed value for the streamflow ($m^3$/s), and $P_i$ is the $i^{th}$ predicted value

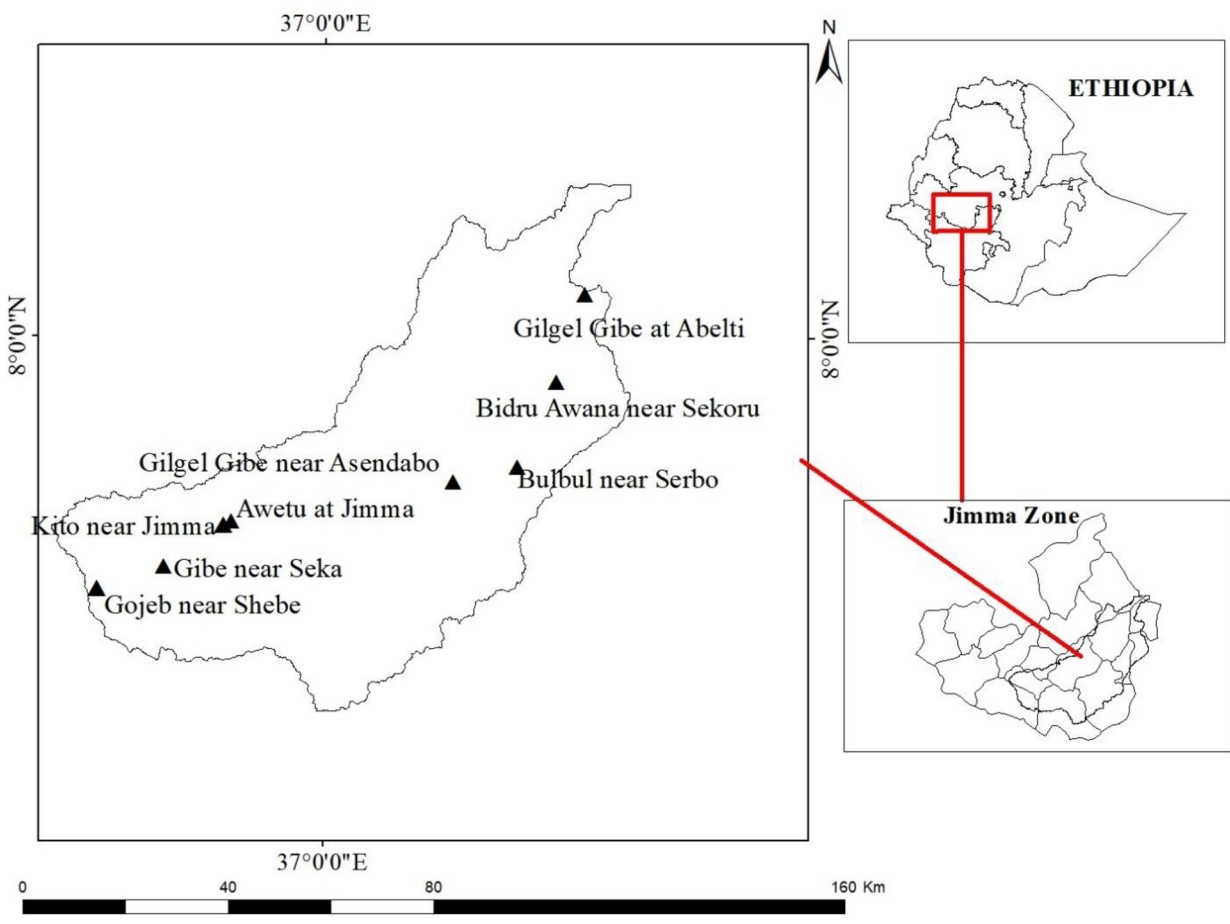

**Fig 9. Gilgel Gibe catchment river discharge gauging stations.**

for the streamflow (m³/s), $\bar{O}$ is the mean of observed streamflow for the entire evaluation time period (m³/s), and $\bar{P}$ is the mean of model predicted stream flow for the entire evaluation time period (m³/s), and n is the total number of observations.

The propagation of uncertainties in model outputs in SUFI-2, expressed as the 95% probability distribution, calculated by the 2.5% and 97.5% levels of the cumulative distributions of output variable is considered as 95PPU [49]. P-factor and R-factor statistics are considered quantifying, the fit between the result expressed as 95PPU and observation. P-factor, the percentage of observations covered by the 95PPU varies from 0 to 1 with the ideal value of 1 while for R-factor, the thickness of the 95PPU optimal value is around 1

## 3. Results and discussion

### 3.1 Hydrological model performance

**3.1.1 Sensitivity analysis, calibration and validation.** Sensitivity analysis for the simulated streamflow was performed using a monthly observed flow to identify the most sensitive parameter with strong influence on model outputs. Initially, parameters related to surface run-off, groundwater, geomorphology, evaporation and soil water were considered and 9 parameters were identified as the most sensitive parameters for calibration. The given ranks are shown in Table 2. The calibration was carried outfitting the model to the streamflow record of

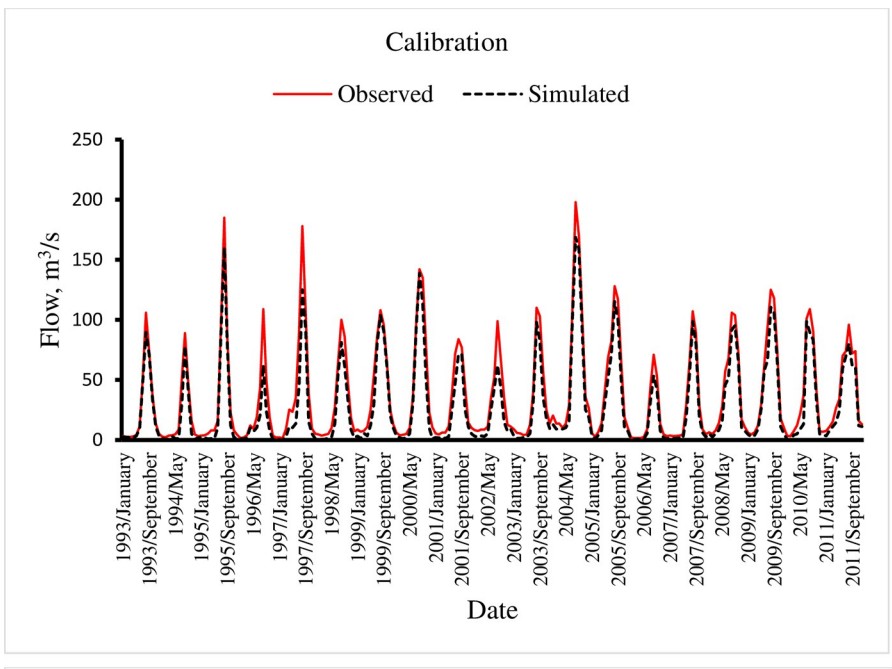

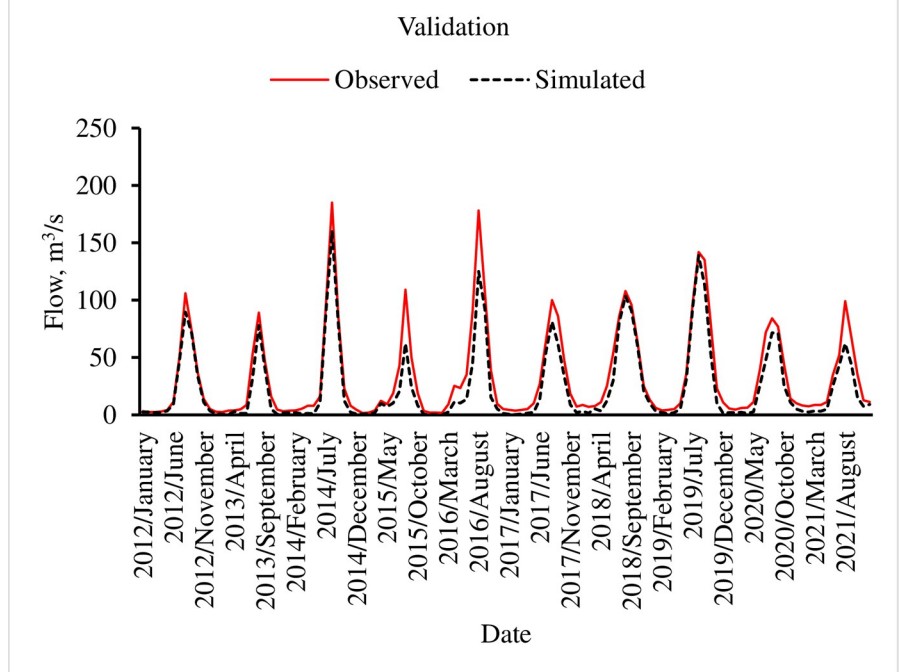

**Fig 10. Calibration and validation of average monthly streamflow.**

1993–2011 and validation was done by streamflow record of 2012–2021. Afterwards, the calibrated values of these 9 parameters were used for further simulations. The parameter ranking was taken from the last iterations of SUFI-2 based on t-stat and p-stat. The larger in the absolute value of t-stat and the smaller the p-value, the more sensitive is the parameter [49]. As it can be seen from Table 2, the top three most sensitive parameters are CN2 (moisture condition II curve number), SOL_AWC (Available water capacity of the soil layer) and RCHRG_DP (Deep aquifer percolation fraction) (Table 2).

**Table 2. Sensitivity analysis and calibrated parameters.**

| No | Parameter Code | Parameter Description | Parameter Value Range | Sensitivity | | |
|----|----------------|-----------------------|------------------------|-------------|---|---|
| | | | | t-stat | p-value | Sensitivity Rank |
| 1 | CN2 | Curve number II | -0.23–0.16 | -17.54 | 0.00 | 1 |
| 2 | SOL_AWC | Available soil water capacity (mm $H_2O$ mm$^{-1}$ soil) | 0–60.26 | -4.04 | 0.00 | 2 |
| 3 | RCHRG_DP | Deep aquifer percolation fraction | -0.04–0.12 | -1.90 | 0.04 | 3 |
| 4 | CH_K2 | Effective hydraulic conductivity in main channel alluvium (mm h$^{-1}$) | 25.18–95.07 | -2.02 | 0.05 | 4 |
| 5 | ALPHA_BF | Base flow recession constant | 0.39–1.08 | 1.61 | 0.07 | 5 |
| 6 | ESCO | Soil evaporation compensation factor | 0.84–1.02 | -1.08 | 0.09 | 6 |
| 7 | CANMX | Maximum canopy storage (mm) | -0.04–0.33 | -3.12 | 0.11 | 7 |
| 8 | SOL_BD | Moist bulk density (Mg/m$^3$) | 0.02–1.06 | 1.38 | 0.13 | 8 |
| 9 | GW_REVAP | Ground water re-evaporation coefficient | 0.08–0.69 | 2.3 | 0.19 | 9 |

The calibration results on mean monthly flow show that SWAT model is able to capture the observed streamflow with $R^2$, NSE, PBIAS and RSR of 0.76, 0.75, 1.02, and 0.49 respectively. The average monthly flow validation indicates $R^2$, NSE, PBIAS and RSR of 0.82, 0.77, 9.4 and 0.47 respectively. This indicates that the performance of the SWAT model in the validation is good enough to simulate the stream flow in the Gilgel Gibe catchment. Furthermore, p-factor and R-factor statistics showed good agreement with 0.87 and 0.79 during calibration and 0.82, and 0.76 during validation, respectively. Overall, the statistics for goodness fit shows good agreement between the observed and simulated average monthly flow.

## 3.2 Climate change projections of the RCMs

The projection of each RCM individually as well as the ensemble mean of the RCMs were examined to show how well each model predicted temperature and precipitation. The distribution mapping of the bias-corrected precipitation and temperature for the future scenarios is compared to the baseline data sets (1991–2021). Further, the hydrological responses of the catchment are analyzed by quantifying and comparing the future water balance components (surface runoff, groundwater flow, total water yield and evapotranspiration).

**3.2.1. Rainfall.** The individual and ensemble mean of the six RCMs were used to assess climate change in two future scenarios; intermediate term (2041–2070) and far future term (2071–2099) with respect to the baseline period (1991–2021) under RCP4.5 and RCP8.5. The results of the projected precipitation by the individual RCMs show a different degree of precipitation changes. Except for HIRHAM5 in far future term, all the RCMs show precipitation decline under both scenarios. According to CCLM4-8, precipitation decreases by -15.87% under RCP4.5 in intermediate term and -30.67% under RCP8.5 scenarios in a far future term.

**Table 3. Changes in precipitation (%) across the four RCMs in RCP4.5 and RCP8.5 scenarios.**

| No | RCMs | RCP4.5 | | RCP8.5 | |
|----|------|--------|---|--------|---|
| | | 2041–2070 | 2071–2099 | 2041–2070 | 2071–2099 |
| 1 | CCLM4-8 | -15.87 | -18.68 | -18.84 | -30.67 |
| 2 | HIRHAM5 | -6.61 | -4.09 | 2.37 | 7.62 |
| 3 | RACMO22T | -12.51 | -20.2 | -9.59 | -7.62 |
| 4 | RCA4 | -10.77 | -15.16 | -13.48 | -14.42 |
| 5 | CRCM5 | -11.66 | -14.32 | -8.69 | -10.22 |
| 6 | REMO2009 | -9.96 | -13.15 | -6.90 | -4.81 |
| | Ensemble | -11.23 | -14.27 | -9.19 | -10.02 |

In case of CRCM5, precipitation decreases by -11.66% under RCP4.5 in intermediate term and -10.22% under RCP8.5 scenarios in a far future term. regarding REMO2009, precipitation decreases by -9.96% under RCP4.5 in intermediate term and -4.81% under RCP8.5 scenarios in a far future term. The highest decline of the precipitation was simulated under RCP8.5 in a far future term with CCLM4-8 and the lowest decline by HIRHAM5 under RCP4.5 in a far future term. HIRHAM5 shows that precipitation will increase under RCP8.5 scenarios.

The ensemble mean of the RCMs suggested decreasing precipitation under the two scenarios, like most of the RCMs. The change shows precipitation decline by -11.23% under RCP4.5 and -9.19% under RCP8.5 in the intermediate term scenario. Under RCP4.5 and RCP8.5, the decline in the far future was -14.27% and -10.02%, respectively (Table 3). The use of ensemble mean over the individual RCMs helps to minimize the highest and lowest projections and enables to minimize the uncertainties by working with the average of the RCMs. Furthermore, the analysis of the inter-annular and inter-model variability of the climate variables in Upper Blue Nile Basin catchment, presented in [53], underlines the need to include several climate models in order to cover the range of possible developments.

Changes in projected precipitation are more profound on seasonal bases compared to the annual bases in the catchment. The projected precipitation shows a declining trend in all seasons, except in April by RCP8.5, for both future time horizons and August by RCP4.5 in an intermediate term scenario. Fig 11 illustrates that the expected changes are higher in the dry season than a wet season. The seasons with lower rainfall (March, April, and May) are likely to experience significant relative changes in precipitation, while the seasons with higher rainfall (June, July, August, and September) are expected to undergo comparatively smaller relative changes. All estimates show a higher declining signal for dry seasons.

Using data from 1965 to 2002, it was shown that the intensity of precipitation was decreasing in Ethiopia's eastern, southwestern, and southern areas [54]. Various studies in the Blue

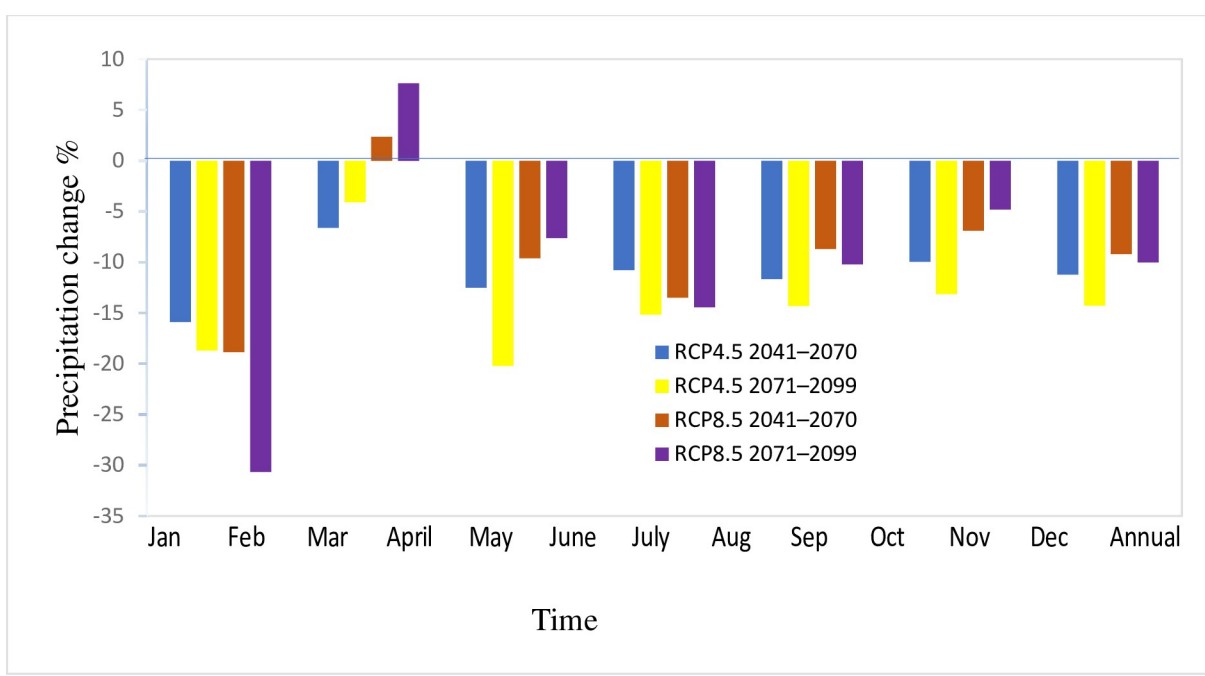

**Fig 11. Percentage change of projected average precipitation in Gilgel Gibe catchment for 2041–2070 and 2071–2099 under RCP4.5 and RCP8.5.**

**Table 4. Changes in temperature (˚C) across the four RCMs in RCP4.5 and RCP8.5 scenarios.**

| No | Tmax | RCP4.5 | | RCP8.5 | |
|---|---|---|---|---|---|
| | RCMs | 2041–2070 | 2071–2099 | 2041–2070 | 2071–2099 |
| 1 | CCLM4-8 | 1.61 | 2.36 | 1.89 | 3.54 |
| 2 | HIRHAM5 | 1.54 | 2.42 | 1.66 | 3.55 |
| 3 | RACMO22T | 1.78 | 2.55 | 1.77 | 3.35 |
| 4 | RCA4 | 1.84 | 2.65 | 2.06 | 3.82 |
| 5 | CRCM5 | 1.64 | 2.44 | 1.77 | 3.48 |
| 6 | REMO2009 | 1.72 | 2.54 | 1.83 | 3.57 |
| | Ensemble | 1.69 | 2.49 | 1.83 | 3.55 |

Nile basin, however, reported varying changes in precipitation in the future periods. According to [55], whose study considered 17 different GCMs, it was found that 10 out of the 17 GCMs reported a decrease in precipitation. However, when the ensemble of all 17 GCMs was analyzed, it revealed essentially no projected change in precipitation. On the other hand, [56] reported an increase in projected precipitation in the late 21st century using 11 GCMs. [57] have also reported significant RCM-GCM uncertainty on the signal of future precipitation in the upper Blue Nile (Abay) river basin. This may be due to the nature of precipitation and its dependence on topographical and physical elements.

The results of precipitation projections in this study are consistent with those of studies on climate change [58, 59] Decreased precipitation was also noted in the study of [60] on the effects of climate change on the hydro-meteorological variables of the Finchaa sub-basin using the HadCM GCM. The study by [61] which also used CCLM downscaling, showed falling precipitation by 6.6% and 6.4% over the upper Blue Nile basin for the periods 2041–2070 and 2071–2100.

**3.2.2 Temperature.** Tables 4 and 5 show the range in the predicted future maximum and minimum temperatures for each individual RCM and their ensemble mean, respectively. The result shows that maximum and minimum temperatures increase under both RCPs throughout the study years considered showing the warming trends in the catchment. However, compared to RCP4.5, the magnitude of changes made by each individual RCM and their mean ensemble is greater for higher emission scenarios (RCP8.5). Under both emission scenarios, the change in temperature is also greater in the far future than the intermediate period. The magnitude of the temperature changes varies across all RCMs and the ensemble mean as well. Under both RCPs for the intermediate and far future, RCA4 has shown the greatest increase in maximum temperature.

Concerning to the ensemble mean of the RCMs, under the medium emission scenario (RCP4.5), the maximum temperature would rise by an average of 1.69˚C and 2.49˚C, while the

**Table 5. Changes in temperature (˚C) across the four RCMs in RCP4.5 and RCP8.5 scenarios.**

| No | Tmin | RCP4.5 | | RCP8.5 | |
|---|---|---|---|---|---|
| | RCMs | 2041–2070 | 2071–2099 | 2041–2070 | 2071–2099 |
| 1 | CCLM4-8 | 1.98 | 3.31 | 2.45 | 5.21 |
| 2 | HIRHAM5 | 1.74 | 2.91 | 1.89 | 4.44 |
| 3 | RACMO22T | 1.58 | 2.59 | 1.94 | 4.05 |
| 4 | RCA4 | 1.52 | 2.38 | 1.93 | 3.73 |
| 5 | CRCM5 | 1.77 | 2.94 | 2.09 | 4.57 |
| 6 | REMO2009 | 1.61 | 2.63 | 1.92 | 4.07 |
| | Ensemble | 1.70 | 2.79 | 2.04 | 4.35 |

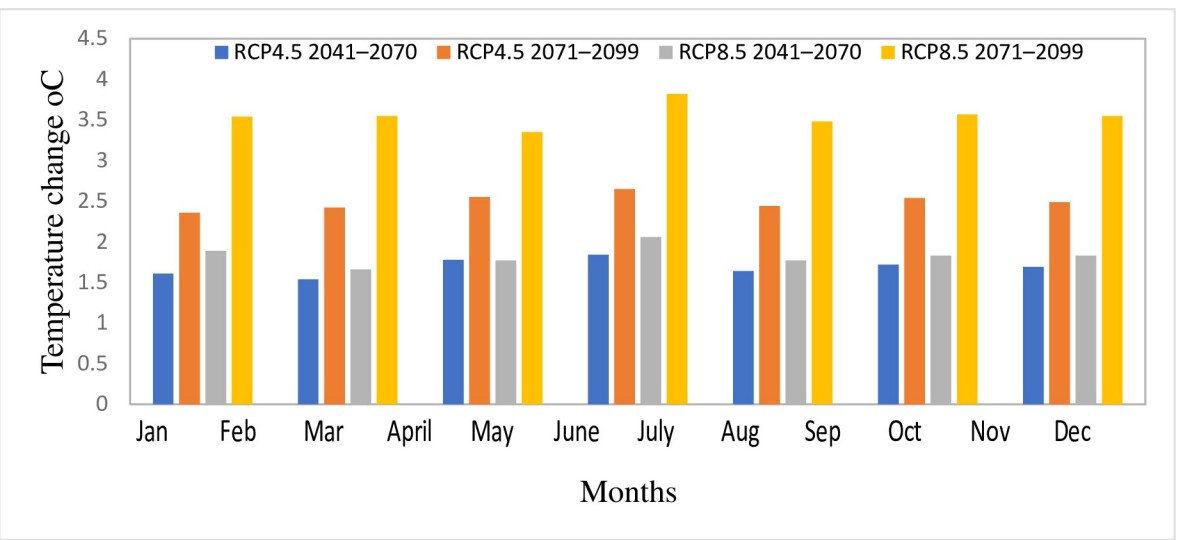

**Fig 12. Seasonal projected change in mean maximum temperature in Gilgel Gibe catchment for 2041–2070 and 2071–2099 under RCP4.5 and RCP8.5.**

minimum temperature will rise by an average of 1.70°C and 2.79°C for the intermediate term and far future, respectively. Under high emission scenarios RCP8.5, the maximum temperature will rise by 1.83°C and 3.55°C, while the minimum temperature would rise by 2.04°C and 4.35°C, respectively, for the intermediate and far future (Table 5). Overall, the analysis showed that RCP8.5's estimate is warmer than RCP4.5's. [62] used five distinct GCMs which confirmed the maximum temperature change in RCP8.5 in the Ilala watershed, Northern Ethiopia. In addition to changing annually, the forecasted temperature changes were also anticipated to vary seasonally. Fig 12 illustrates that the seasonal changes are greater for the minimum temperature than the maximum temperature. Maximum temperature fluctuations are more pronounced during dry seasons (December, January, and February), while minimum temperature changes are more pronounced during wet seasons (June, July, and August) (Fig 13). The highest seasonal rise of maximum temperature is reported by RCP8.5 during 2071–2099 while RCP 4.5 reported the lowest rise during 2041–2070.

Similar to annual fluctuations, the high emission scenario is anticipated to cause a greater seasonal rise in temperature than the medium emission scenario. In addition, the far future scenario (2071–2099) is anticipated to have higher increases in maximum and minimum temperatures than the intermediate future scenario (2041–2070).

Overall, the forecast of the maximum temperature and minimum temperature over both future time horizons are consistent with other researchers' ranges and falls within the IPCC's range. Numerous research on climate change in Ethiopia have suggested that temperatures are likely to rise across the country's various regions. However, the degree of change varies according on the downscaling methods and kinds of climate models. According to the Global Facility for Disaster Reduction and Recovery [63] analysis on the country profile for climate risk and adaptation, the mean annual temperature is expected to rise by 1.1 to 3.1 degrees Celsius by the 2060s and by 1.5 to 5.1 degrees Celsius by the 2090s. According to the analysis by [61], the entire upper Blue Nile basin's average annual temperature will rise by 1.5°C, 2.6°C, and 4.5°C between the years 2011 and 2040, 2041 and 2070, and 2071 and 2100, respectively. According to [64] study's on the entire Nile Basin, which used an average of 11 GCMs, the yearly average temperature change between 1950 and 1999 and 2010 to 2039 will range from 0.91 to 1.9

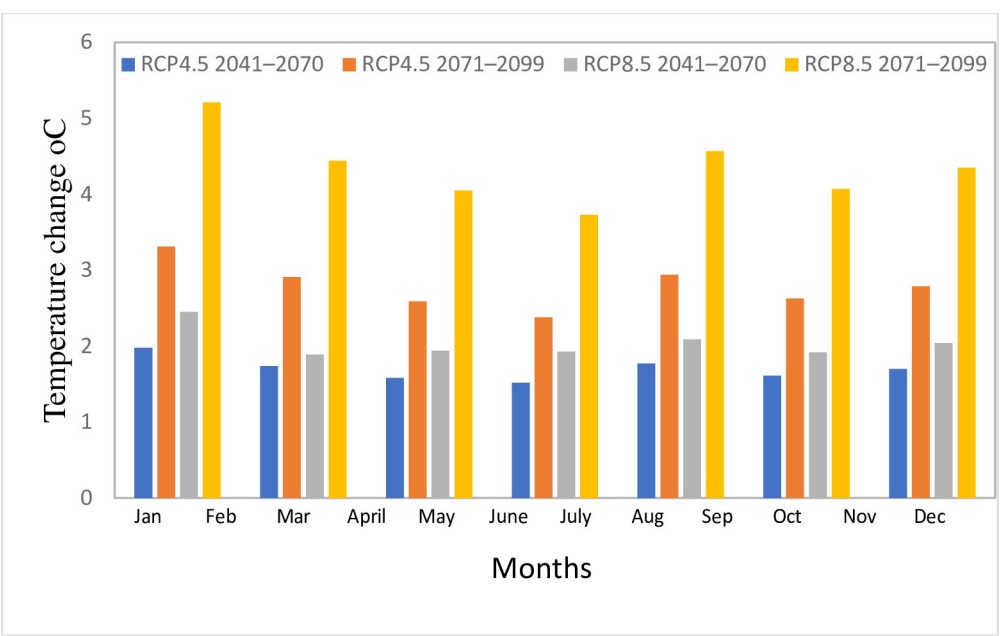

**Fig 13. Seasonal projected change in mean minimum temperature in Gilgel Gibe catchment for 2041–2070 and 2071–2099 under RCP4.5 and RCP8.5.**

degrees Celsius. Furthermore, [57].'s study on the Blue Nile, which used 17 GCMs, predicted that temperatures will rise by 2 ˚C and 5 ˚C.

**3.2.3 Impacts of climate change.**   Changes in rainfall and temperature were used to predict the impacts of climate change on water balance components of the catchment. Consequently, the projected changes in mean annual precipitation and temperature, under the two RCPs, caused a significant variation in the projected water balance components of Gilgel Gibe catchment.

Reduced surface flow, groundwater, and overall water yield are predicted as a result of the decline in precipitation and rise in temperature, according to SWAT models for the intermediate term and far future Table 6. Changes in temperature and PET are correlated positively. In turn, this led to a probable rise in evapotranspiration and evaporation, which may have a significant impact on the decline in water yield. Evaporation is a factor in the future decline of the surface runoff since the forms of water are prone to losses due to temperature changes. This is expected considering the warming trends of temperature.

Generally, the interactions of precipitation, temperature, and evapotranspiration can be used to describe how climate change has affected the hydrology of a catchment. Reduced rainfall lowers surface runoff, while rising temperatures enhanced an increase in evapotranspiration that would have caused the runoff to decline. In this regard, the correlation between precipitation and the simulated surface runoff, groundwater and total water yield is positive, while the relationship between the temperature and hydrological process are positive only for PET. However, the relationship between the precipitation and temperature with the hydrological process varies with the RCPs and period of the scenario. Changes in precipitation, the expected decline in groundwater and total water yields, for instance, is larger under both RCPs for the period 2071–2100. However, under RCP4.5, the reduction in surface runoff is only more pronounced during the years 2071–2099 (-14.71%) (Fig 14). Fig 11 illustrates how the decline in surface water under RCP8.5 is greater throughout the period 2040–2070 (-12.55%).

**Table 6. Changes of water balance components under a climate change.**

| RCM | RCP | Time Period | Surface Runoff (%) | Groundwater flow (%) | Water Yield (%) | PET (%) |
|---|---|---|---|---|---|---|
| CCLM4-8 | RCP4.5 | 2041–2070 | -7.61 | -9.34 | -8.92 | 16.01 |
| CCLM4-8 | RCP4.5 | 2071–2099 | -14.96 | -13.83 | -14.01 | 19.02 |
| CCLM4-8 | RCP8.5 | 2041–2070 | -12.65 | -9.89 | -10.58 | 16.69 |
| CCLM4-8 | RCP8.5 | 2071–2099 | -9.57 | -16.08 | -13.76 | 22.67 |
| HIRHAM5 | RCP4.5 | 2041–2070 | -7.79 | -9.32 | -8.78 | 16.15 |
| HIRHAM5 | RCP4.5 | 2071–2099 | -14.64 | -13.98 | -14.03 | 19 |
| HIRHAM5 | RCP8.5 | 2041–2070 | -13.87 | -9.91 | -10.99 | 16.79 |
| HIRHAM5 | RCP8.5 | 2071–2099 | -9.96 | -16.02 | -13.94 | 22.78 |
| RACMO22T | RCP4.5 | 2041–2070 | -7.14 | -9.11 | -8.19 | 16.05 |
| RACMO22T | RCP4.5 | 2071–2099 | -14.59 | -13.99 | -14.04 | 19.01 |
| RACMO22T | RCP8.5 | 2041–2070 | -13.72 | -9.48 | -10.31 | 16.68 |
| RACMO22T | RCP8.5 | 2071–2099 | -9.31 | -16.27 | -13.51 | 22.69 |
| RCA4 | RCP4.5 | 2041–2070 | -7.46 | -9.73 | -8.89 | 16.12 |
| RCA4 | RCP4.5 | 2071–2099 | -14.85 | -13.91 | -14.02 | 19.02 |
| RCA4 | RCP8.5 | 2041–2070 | -13.67 | -9.63 | -10.05 | 16.48 |
| RCA4 | RCP8.5 | 2071–2099 | -9.34 | -16.03 | -13.71 | 22.47 |
| CRCM5 | RCP4.5 | 2041–2070 | -7.84 | -9.76 | -8.82 | 16.07 |
| CRCM5 | RCP4.5 | 2071–2099 | -14.48 | -13.96 | -14.07 | 19.01 |
| CRCM5 | RCP8.5 | 2041–2070 | -13.92 | -9.99 | -10.82 | 16.75 |
| CRCM5 | RCP8.5 | 2071–2099 | -9.51 | -16.05 | -13.78 | 22.69 |
| REMO2009 | RCP4.5 | 2041–2070 | -7.55 | -9.45 | -8.73 | 16.09 |
| REMO2009 | RCP4.5 | 2071–2099 | -14.72 | -13.92 | -14.01 | 19.02 |
| REMO2009 | RCP8.5 | 2041–2070 | -12.56 | -9.78 | -10.54 | 16.68 |
| REMO2009 | RCP8.5 | 2071–2099 | -9.52 | -16.08 | -13.75 | 22.67 |
| Ensemble | RCP4.5 | 2041–2070 | -7.56 | -9.44 | -8.72 | 16.08 |
| Ensemble | RCP4.5 | 2071–2099 | -14.71 | -13.93 | -14 | 19.01 |
| Ensemble | RCP8.5 | 2041–2070 | -12.55 | -9.79 | -10.55 | 16.67 |
| Ensemble | RCP8.5 | 2071–2099 | -9.53 | -16.09 | -13.74 | 22.66 |

This demonstrates that changes in precipitation is not the only factor affecting surface water. This result was in line with previous studies on the Nile River. In the upper Nile, [65] found that hot, dry years are occurring more frequently due to rising regional temperatures.

The spatial distribution of the hydrological process under the projected climate change for RCP4.5 and RCP8.5 is presented in the Figures. The catchment's surface runoff, groundwater, and water yield were all reported to have decreased due to climate change, but the yields in each sub-basin varied. While temperature changes are projected to have indirect effects, spatiotemporal patterns and magnitudes of precipitation are predicted to directly disrupt the hydrological process. In both scenarios throughout the study projections taken into consideration, the upstream of the catchment is often anticipated to have the greatest decline in surface runoff, groundwater, and water yield. Under the RCP4.5 and RCP8.5 scenarios, the projection for the decline in surface runoff and water yield is greater in the intermediate than in the far future. In both scenarios, the southwest is where surface runoff is expected to decline at the slowest rate over the intermediate term and the far future, followed by the northeast. The southwest corner has a higher rate of groundwater and water yield decline. The highest rise in PET is shown by RCP8.5 around the outlet of the catchment in the mid future projection.

In both scenarios, it was observed that the catchment's surface runoff had decreased as a result of climate change. The upstream of the catchment is frequently expected to have the

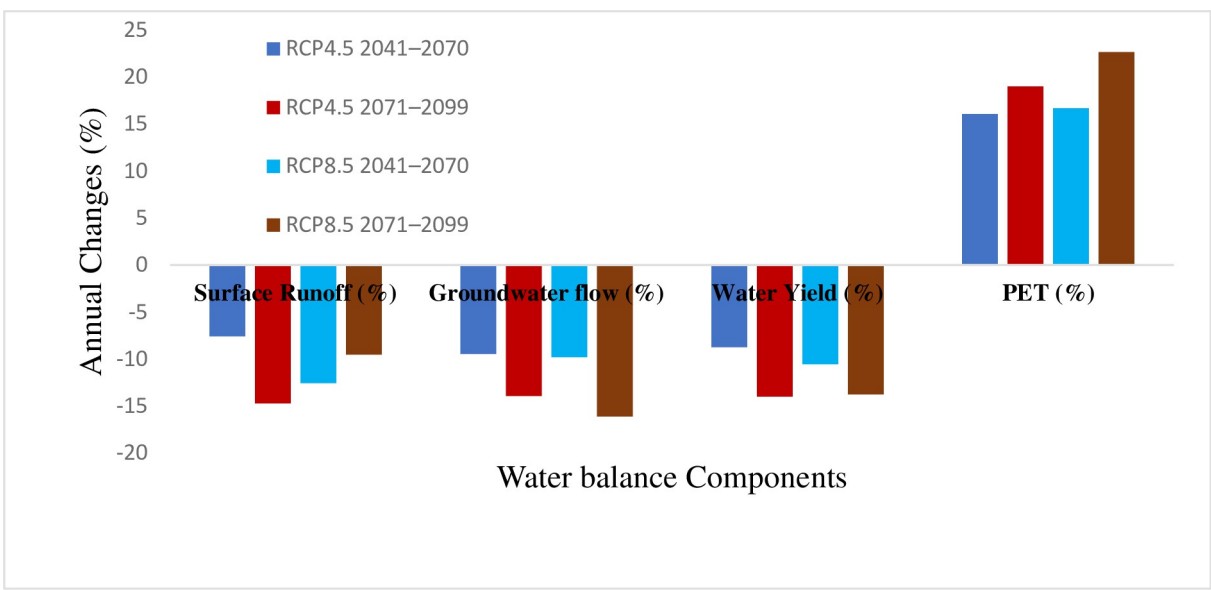

**Fig 14. Impacts of climate change on water balance.**

highest decline in surface runoff in both scenarios considered in the study forecasts. The projected decline in surface runoff is greater in the intermediate term than in the far future, according to the RCP4.5 and RCP8.5 scenarios. Surface runoff is anticipated to decrease at the slowest rate in both scenarios over the far future and the intermediate term in the southwest, followed by the northeast. Reduced surface runoff in the catchment are all potential outcomes that could affect the catchment's water resource availability and exacerbate water shortages (Fig 15).

According to the result, climate change has caused a decline in the catchment's groundwater flow. The upstream of the catchment is frequently expected to have the biggest groundwater decline in all scenarios that take into account study forecasts. Groundwater levels are declining faster in the southwest corner. Water stress downstream could be made worse by the potential for lower groundwater levels in the catchment, which would affect the availability of water resources in the catchment (Fig 16).

The upstream of the catchment is expected to have the highest decrease in water yield under both scenarios considered in the study estimates. The projected loss in water yield is larger in the intermediate than in the far future, according to the RCP4.5 and RCP8.5 scenarios. The rate of water yield decline is greater in the southwest corner. The potential for a decreased total water yield in the catchment could affect the availability of water resources in the catchment and worsen water stress downstream (Fig 17).

In general, the increase in temperature and decline in annual and seasonal precipitation in both scenarios could be the cause of the catchment's decreased average annual flow. The possibility of reduced surface water, groundwater, and total water yield in the catchment could have an impact on the availability of water resources in the catchment and exacerbate water stress downstream. The acceleration of the increasing water abstractions could also contribute to the further decline in water resources besides the climate change pressure.

Reduced soil moisture, which is necessary for plant growth and groundwater storage, may result from the increase in PET (Fig 18) brought on by rising temperatures and a decline in precipitation. This may indicate decreased water availability for agricultural production, which will become a chronic problem for the farmers in the catchment whose only source of

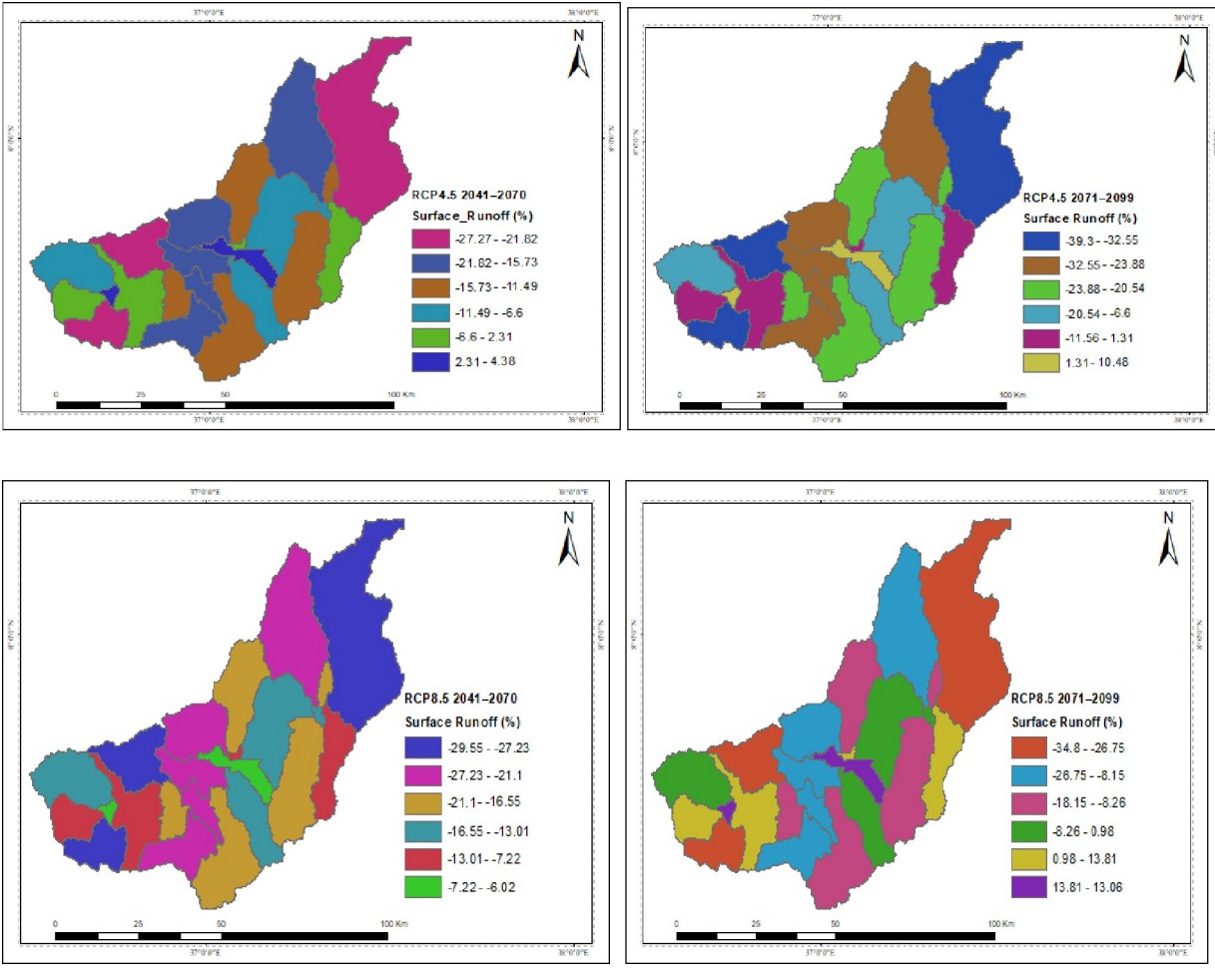

**Fig 15. Spatial patterns of the changes in surface runoff due to climate change.**

income is agriculture. Additionally, a rise in temperature increased PET, which in turn raised the requirement for irrigation.

Similar findings were supported by research on climate change in Africa, namely the sub-Sahara, which showed a range of warming trends in the inland subtropics, a recurrence of high hot events, an increase in aridity, and changes in rainfall [66]. The effects of climate change, according to [66], have a considerable impact on the region's undernourishment, infectious diseases, vulnerability to rain-fed agriculture, and flash flooding catastrophes.

The results also support the findings of [57], which indicated that the Nile river is predicted to decline between the years of 2040 and 2070 and again between 2070 and 2099 as a result of decreased precipitation and higher evaporation rate. In their study on the effects of climate change on the Tekeze river basin's water sources, [67] also predicted that surface runoff would decrease by 13% and 14%. The decline of the expected surface runoff under both RCPs due to climate change was also confirmed in the study by [62] using the ensemble of five GCMs.

## 4. Conclusion and management options

This study undertook an analysis of the effects of climate change on the hydrological processes within the Gilgel Gibe catchment. Specifically, the study evaluated the impact of potential

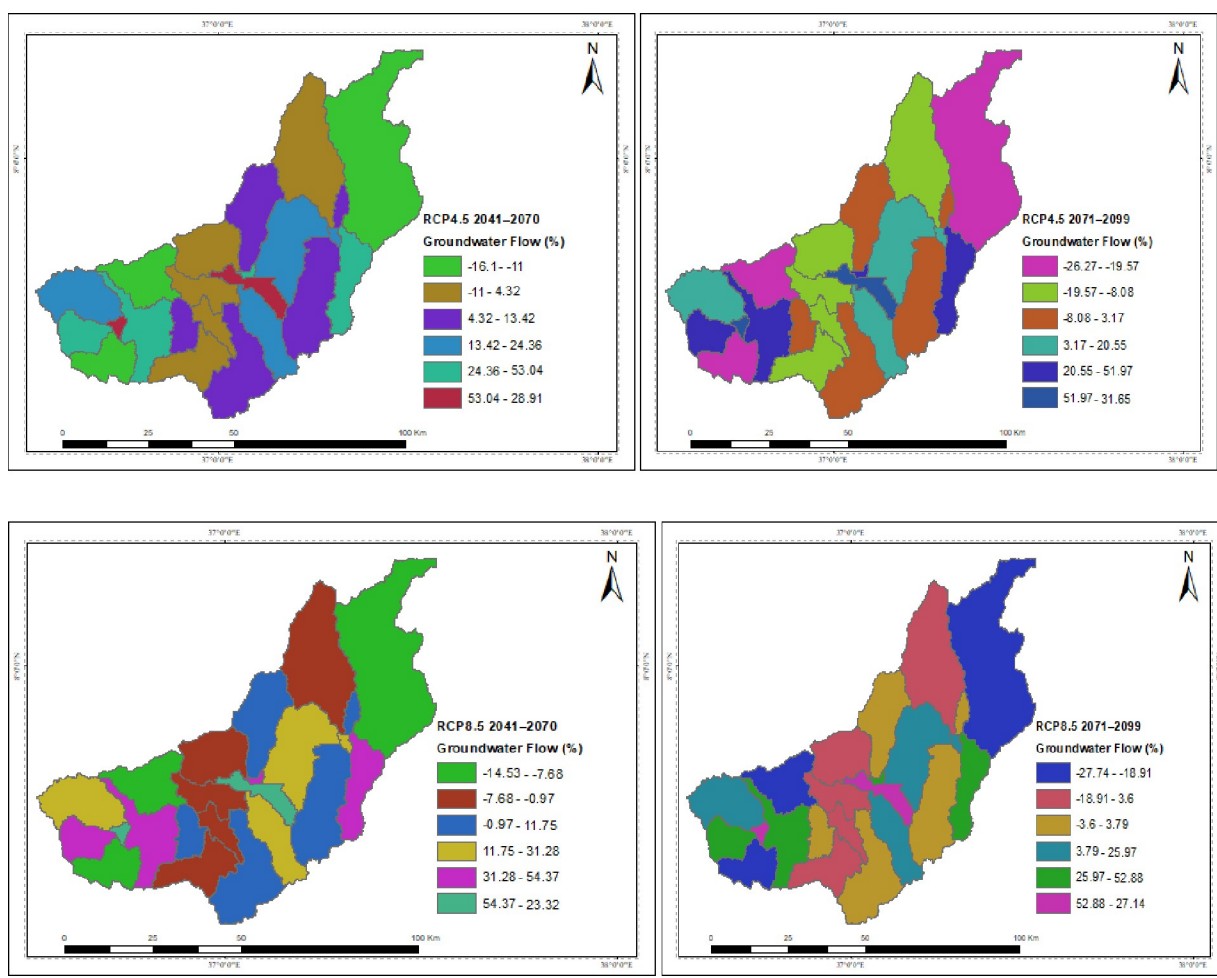

**Fig 16. Spatial patterns of the changes in groundwater due to climate change.**

future climate changes using six Global Climate Models (GCMs), and their corresponding ensemble mean, through the application of a physically based distributed hydrologic model known as Soil and Water Assessment Tool (SWAT). The study utilized six Regional Climate Models (RCMs), and their ensemble mean, from the Coordinated Regional Climate Downscaling Experiment for Africa (CORDEX-Africa) to examine the potential implications of climate change on the Gilgel Gibe catchment. Notably, the RCM simulations indicated a decrease in precipitation levels for scenarios involving high and medium-low emissions, with the exception of HIRHAM5. In terms of temperature, all RCM projections indicate an upward trend with varying magnitudes of change. To analyze the hydrological impacts of climate change, the ensemble mean of the RCMs was employed to mitigate the substantial variability inherent in precipitation and temperature forecasts by different RCMs. The use of ensemble RCMs was found to be a suitable approach for evaluating uncertainties in individual RCMs, despite the presence of inherent uncertainties in climate prediction. As a result of the anticipated decrease in precipitation and the increase in temperature, surface runoff, groundwater, and total water yield are all anticipated to decrease, while potential evapotranspiration is expected to increase. Overall, the water resources of the Gilgel Gibe catchment are exceedingly sensitive to changes in climate.

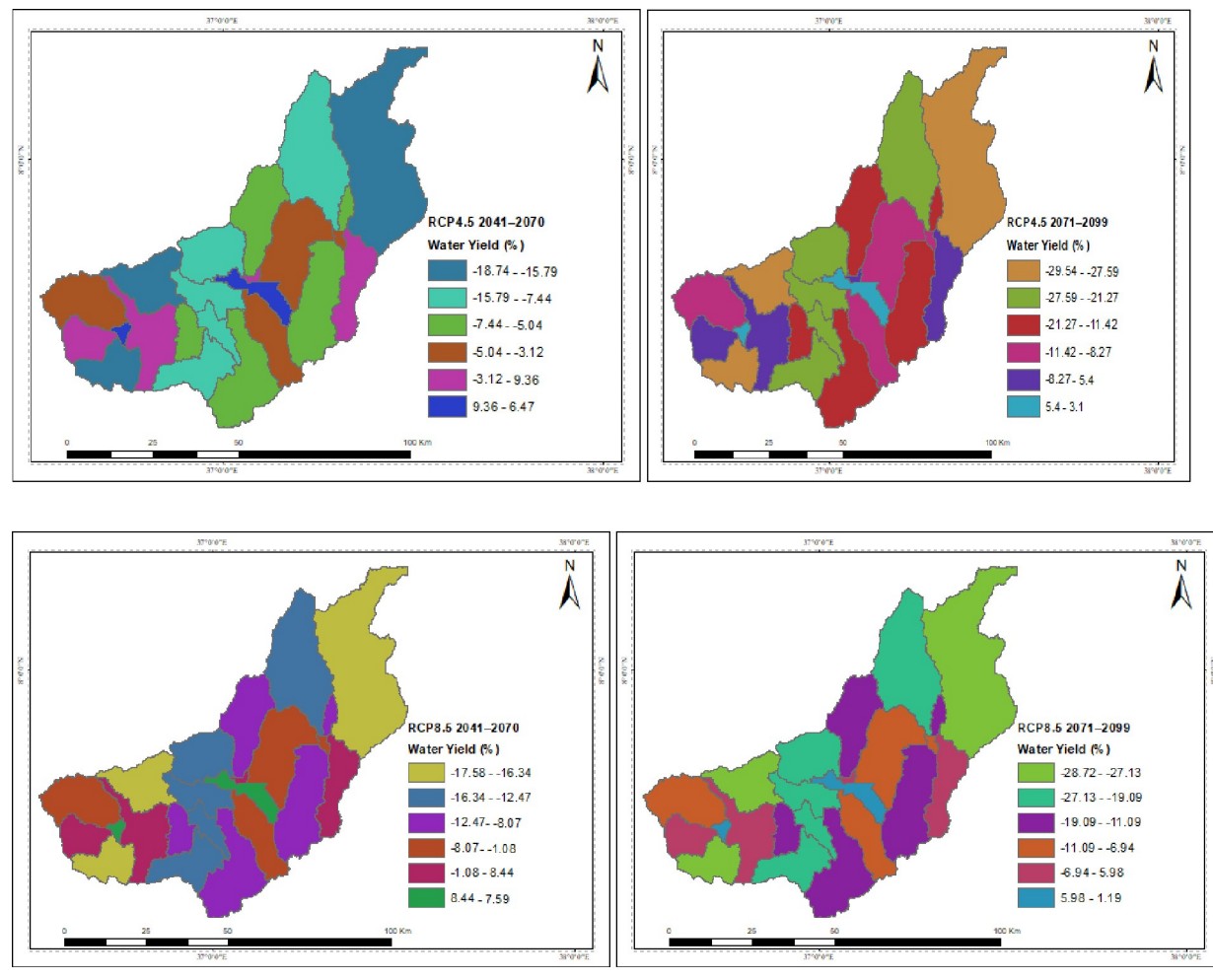

**Fig 17. Spatial patterns of the changes in water yield due to climate change.**

Furthermore, the Gilgel Gibe catchment exhibits significant temporal and spatial variability in temperature and precipitation, which manifests in heightened occurrences of hot and dry years, leading to severe water scarcity. Consequently, the catchment is highly susceptible to the impacts of climate change, such as warming and crop failures resulting from prolonged dry seasons. The observed decrease in runoff within the catchment has contributed to enhanced water security. The study's analysis indicates that the SWAT simulations provide adequate estimates of the impacts of climate change in the Gilgel Gibe catchments. To enhance our understanding of the present and future climate dynamics, urgent attention is required to improve the availability and quality of hydro-climatic data in the region.

The results of this study have provided crucial insights into the impacts of climate change on the hydrological processes of the Gilgel Gibe catchment. These findings can inform the development of effective water resource management initiatives in the region. The rehabilitation of degraded sloping lands could enhance ground recharge and reduce surface runoff that contributes to soil erosion and sedimentation in the Gilgel Gibe I dam. Additionally, appropriate management strategies should be implemented to safeguard the Gilgel Gibe I hydroelectric dam in the catchment. Overall, these results underscore the imperative for regional development and collaboration to facilitate robust and climate-resilient management strategies and mitigate the adverse effects of climate change on the catchment.

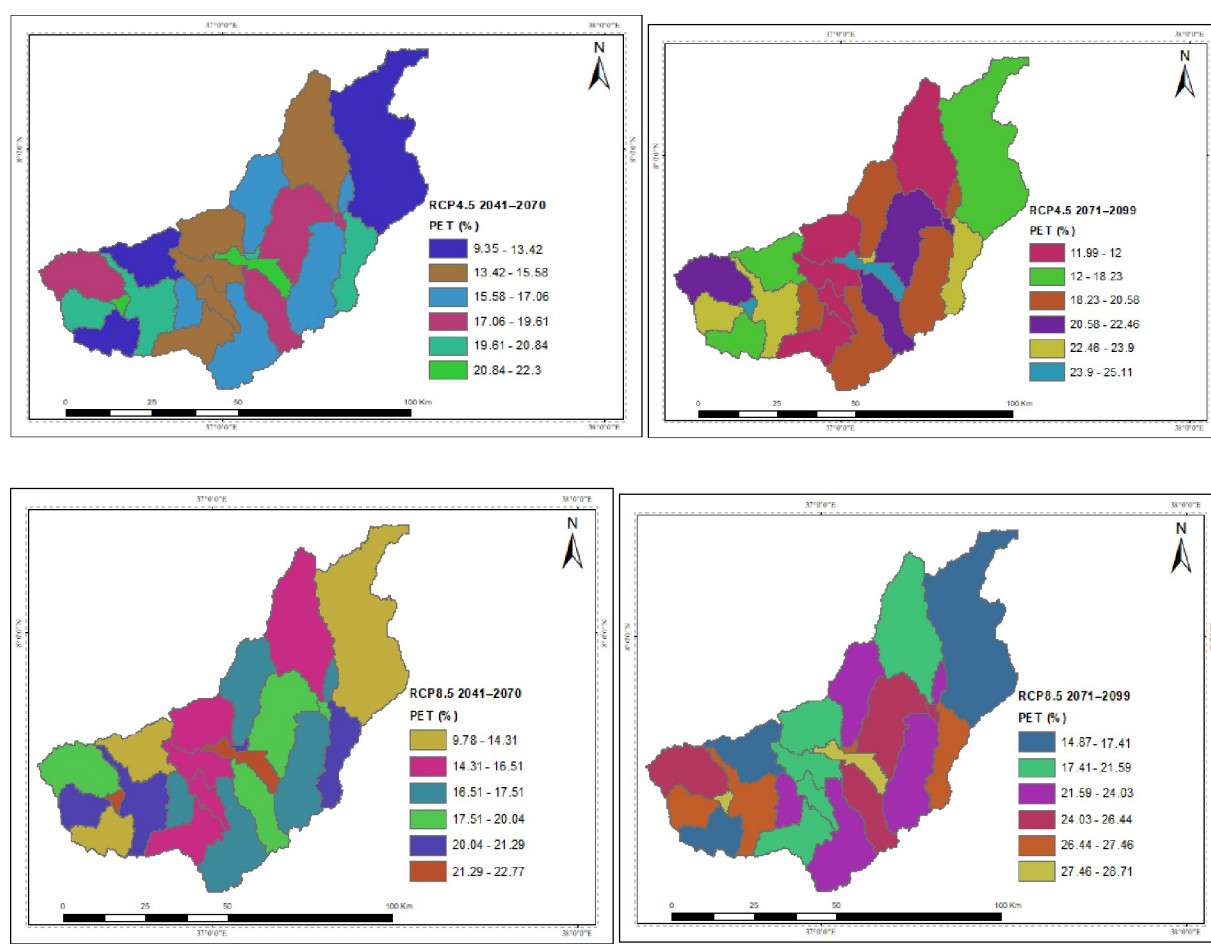

**Fig 18. Spatial patterns of the changes in Potential Evapotranspiration (PET) due to climate change.**

## Author Contributions

**Conceptualization:** Zewde Alemayehu Tilahun, Yechale Kebede Bizuneh, Abren Gelaw Mekonnen.

**Data curation:** Zewde Alemayehu Tilahun, Yechale Kebede Bizuneh, Abren Gelaw Mekonnen.

**Formal analysis:** Zewde Alemayehu Tilahun.

**Investigation:** Zewde Alemayehu Tilahun, Abren Gelaw Mekonnen.

**Methodology:** Zewde Alemayehu Tilahun, Abren Gelaw Mekonnen.

**Software:** Zewde Alemayehu Tilahun, Abren Gelaw Mekonnen.

**Supervision:** Yechale Kebede Bizuneh, Abren Gelaw Mekonnen.

**Validation:** Yechale Kebede Bizuneh, Abren Gelaw Mekonnen.

**Visualization:** Abren Gelaw Mekonnen.

**Writing – original draft:** Zewde Alemayehu Tilahun.

**Writing – review & editing:** Zewde Alemayehu Tilahun, Yechale Kebede Bizuneh, Abren Gelaw Mekonnen.

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
