## [Decision Letter · Decision Letter 0]

20 Feb 2023

PONE-D-22-23861The Impacts of Climate Change on Hydrological Processes of Gilgel Gibe Catchment, Southwest EthiopiaPLOS ONE

Dear Dr. Tilahun,

Thank you for submitting your manuscript to PLOS ONE. After careful consideration, we feel that it has merit but does not fully meet PLOS ONE’s publication criteria as it currently stands. Therefore, we invite you to submit a revised version of the manuscript that addresses the points raised during the review process.

We look forward to receiving your revised manuscript.

Kind regards,

Salim Heddam

Academic Editor

PLOS ONE

2.We note that you have stated that you will provide repository information for your data at acceptance. Should your manuscript be accepted for publication, we will hold it until you provide the relevant accession numbers or DOIs necessary to access your data. If you wish to make changes to your Data Availability statement, please describe these changes in your cover letter and we will update your Data Availability statement to reflect the information you provide.

3. Please include a caption for figure 1-4, 8-9,13 and 17.

4. We note that [Figures 1, 2, 6, 7, 8, 14, 15, 16 and 17] in your submission contain [map/satellite] images which may be copyrighted. All PLOS content is published under the Creative Commons Attribution License (CC BY 4.0), which means that the manuscript, images, and Supporting Information files will be freely available online, and any third party is permitted to access, download, copy, distribute, and use these materials in any way, even commercially, with proper attribution. For these reasons, we cannot publish previously copyrighted maps or satellite images created using proprietary data, such as Google software (Google Maps, Street View, and Earth). For more information, see our copyright guidelines: http://journals.plos.org/plosone/s/licenses-and-copyright.

a.You may seek permission from the original copyright holder of [Figures 1, 2, 6, 7, 8, 14, 15, 16 and 17] to publish the content specifically under the CC BY 4.0 license.   

Natural Earth (public domain): http://www.naturalearthdata.com/.

Additional Editor Comments:

Reviewer 1

The work, " The Impacts of Climate Change on Hydrological Processes of Gilgel Gibe Catchment, Southwest Ethiopia" assesses the changes in hydrological parameters due to changing climate in the region. Overall the work is good. However, more exhaustive conclusion is further required to relate the changing climate impacts on hydrological processes and other forcing factors operating in the region.

Some of the specific comments are listed below:

Line 40 rephrase the sentence or rewrite the whole first paragraph. Authors had loosely written introduction. The main function of an introduction section is to contextualise your study, here it is somewhat missing.

Line 87, rephrase the beginning sentence of paragraph?

Line 109-114, Authors need to check last paragraph of introduction, please formulate explicitly your research questions and the novelty of your work and state clearly the specific paper objectives. Authors are requested go through results and conclusion thoroughly first and then write last paragraph of introduction.

Line 179-180, Authors have chosen Distribution mapping (DM) method for bias correction for both precipitation and temperature based on absolute error ranking. However, authors had discussed the studies where DM method have been used. It is suggested that the authors could have provided table in which absolute error value for other methods (Linear scaling (multiplicative), delta-change correction (multiplicative), precipitation local intensity scaling, and power transformation of precipitation with respect to DM) for Precipitation data and (Linear scaling (additive), delta-change correction (additive), variance scaling of temperature with respect to DM) for temperature data are provided. At least provide 2 methods for comparison.

Line 185-186, rephrase the sentence “Based on mean absolute error ranking, distribution mapping was ranked both, for temperature and precipitation corrections”.

Line 241-247, Landsat Thematic Mapper of 1991, Enhanced Thematic Mapper Plus of 2006, and Operational Land Imager or Thermal Infrared Sensor of 2021 were used for LULC classifications. Which LULC map is used for SWAT Model? Authors had employed different satellite data but didn’t mention LULC map used in the study. Please make it clear.

Line 257, Authors need to mention metrological station names only in place of their coordinates or put a table for name and coordinates of Met. Stations.

Line 266, Authors are advised either write all weather data used in the SWAT for simulation with warmup periods, calibration and validation or remove this sentence. Moreover, weather data from 1991 and 1992 were used as the model initialization phase (warm up) and not included in the final analysis.

Line 268, Same comment as of 257

Authors are advised check this paragraph “Sensitivity Analysis, Calibration and Validation”, it is misleading. Authors had performed analysis on daily observed flow or mean monthly flow, please make it clear.

Line 320-321, Sensitivity analysis for the simulated streamflow was performed using a daily observed flow to identify the most sensitive parameter with strong influence on model outputs.

Line 334-335 The calibration results on mean monthly flow show that SWAT model is able to capture the observed streamflow with R2, NSE, PBIAS and RSR of 0.76, 0.75, 1.02, and 0.49 respectively.

Line 530, Conclusion is not conclusive, make it clear particularly last two paragraphs of conclusion.

Figures, give appropriate number to figures, both in text and at the end.

Authors are suggested to modify the flow chat, there are many horrible mistakes in it.

Reviewer 2

Well conceived and executed. Study findings are very important to understand the impact of environments on the water resource. Abstract needs some clarity. Introduction should be concise. English of the manuscript needs to work out.

Reviewer 3

The paper by Tilahun et al deals with climate changes in the Gilgel Gibe Catchment and related impacts on the hydrologic processes. I believe the paper has potential to be published in Plos-One because the methodology followed by the authors is generally rigorous and results are of interest for the region. However, it needs a substantial revision (analysis of uncertainties, form).

1) Climate change analysis is carried out based on 6 RCMs. Hydrological impacts are calculated only on the ensemble mean. I believe it should be analyzed how uncertainties in climate reflect on the hydrological processes. A run of the hydrological model with the ensemble mean forcing is ok, but it should be accompanied by runs forced with every RCM inputs. Science delas with uncertainties and these uncertainties must be calculated and communicated especially in studies that want to be a reference for environmental management.

2) The manuscript is full of repetitions. It needs to be carefully proofread for conciseness.

3) There are many single figures, I believe some could be grouped in multi-panel figures.

4) English must be reviewed, there are many inaccuracies in the use of verbal forms (past/present; singular/plural), articles and so on.

5) Authors mentioned that data are fully available but there is no information about the repository where they can be found.

6) I have attached a commented manuscript in which every general comment above (except 5) is backed up by detailed comments and suggestions.

Reviewers' comments:

Reviewer's Responses to Questions

**Comments to the Author**

1. Is the manuscript technically sound, and do the data support the conclusions?

Reviewer #1: Yes

Reviewer #2: Yes

Reviewer #3: Partly

2. Has the statistical analysis been performed appropriately and rigorously? 

Reviewer #1: Yes

Reviewer #2: Yes

Reviewer #3: Yes

3. Have the authors made all data underlying the findings in their manuscript fully available?

Reviewer #1: Yes

Reviewer #2: Yes

Reviewer #3: Yes

4. Is the manuscript presented in an intelligible fashion and written in standard English?

Reviewer #1: Yes

Reviewer #2: No

Reviewer #3: No

5. Review Comments to the Author

Reviewer #1: The work, " The Impacts of Climate Change on Hydrological Processes of Gilgel Gibe Catchment, Southwest Ethiopia" assesses the changes in hydrological parameters due to changing climate in the region. Overall the work is good. However, more exhaustive conclusion is further required to relate the changing climate impacts on hydrological processes and other forcing factors operating in the region.

Some of the specific comments are listed below:

Line 40 rephrase the sentence or rewrite the whole first paragraph. Authors had loosely written introduction. The main function of an introduction section is to contextualise your study, here it is somewhat missing.

Line 87, rephrase the beginning sentence of paragraph?

Line 109-114, Authors need to check last paragraph of introduction, please formulate explicitly your research questions and the novelty of your work and state clearly the specific paper objectives. Authors are requested go through results and conclusion thoroughly first and then write last paragraph of introduction.

Line 179-180, Authors have chosen Distribution mapping (DM) method for bias correction for both precipitation and temperature based on absolute error ranking. However, authors had discussed the studies where DM method have been used. It is suggested that the authors could have provided table in which absolute error value for other methods (Linear scaling (multiplicative), delta-change correction (multiplicative), precipitation local intensity scaling, and power transformation of precipitation with respect to DM) for Precipitation data and (Linear scaling (additive), delta-change correction (additive), variance scaling of temperature with respect to DM) for temperature data are provided. At least provide 2 methods for comparison.

Line 185-186, rephrase the sentence “Based on mean absolute error ranking, distribution mapping was ranked both, for temperature and precipitation corrections”.

Line 241-247, Landsat Thematic Mapper of 1991, Enhanced Thematic Mapper Plus of 2006, and Operational Land Imager or Thermal Infrared Sensor of 2021 were used for LULC classifications. Which LULC map is used for SWAT Model? Authors had employed different satellite data but didn’t mention LULC map used in the study. Please make it clear.

Line 257, Authors need to mention metrological station names only in place of their coordinates or put a table for name and coordinates of Met. Stations.

Line 266, Authors are advised either write all weather data used in the SWAT for simulation with warmup periods, calibration and validation or remove this sentence. Moreover, weather data from 1991 and 1992 were used as the model initialization phase (warm up) and not included in the final analysis.

Line 268, Same comment as of 257

Authors are advised check this paragraph “Sensitivity Analysis, Calibration and Validation”, it is misleading. Authors had performed analysis on daily observed flow or mean monthly flow, please make it clear.

Line 320-321, Sensitivity analysis for the simulated streamflow was performed using a daily observed flow to identify the most sensitive parameter with strong influence on model outputs.

Line 334-335 The calibration results on mean monthly flow show that SWAT model is able to capture the observed streamflow with R2, NSE, PBIAS and RSR of 0.76, 0.75, 1.02, and 0.49 respectively.

Line 530, Conclusion is not conclusive, make it clear particularly last two paragraphs of conclusion.

Figures, give appropriate number to figures, both in text and at the end.

Authors are suggested to modify the flow chat, there are many horrible mistakes in it.

Reviewer #2: Well conceived and executed. Study findings are very important to understand the impact of environments on the water resource. Abstract needs some clarity. Introduction should be concise. English of the manuscript needs to work out.

Reviewer #3: The paper by Tilahun et al deals with climate changes in the Gilgel Gibe Catchment and related impacts on the hydrologic processes. I believe the paper has potential to be published in Plos-One because the methodology followed by the authors is generally rigorous and results are of interest for the region. However, it needs a substantial revision (analysis of uncertainties, form).

1) Climate change analysis is carried out based on 6 RCMs. Hydrological impacts are calculated only on the ensemble mean. I believe it should be analyzed how uncertainties in climate reflect on the hydrological processes. A run of the hydrological model with the ensemble mean forcing is ok, but it should be accompanied by runs forced with every RCM inputs. Science delas with uncertainties and these uncertainties must be calculated and communicated especially in studies that want to be a reference for environmental management.

2) The manuscript is full of repetitions. It needs to be carefully proofread for conciseness.

3) There are many single figures, I believe some could be grouped in multi-panel figures.

4) English must be reviewed, there are many inaccuracies in the use of verbal forms (past/present; singular/plural), articles and so on.

5) Authors mentioned that data are fully available but there is no information about the repository where they can be found.

6) I have attached a commented manuscript in which every general comment above (except 5) is backed up by detailed comments and suggestions.

6. PLOS authors have the option to publish the peer review history of their article (what does this mean?). If published, this will include your full peer review and any attached files.

Reviewer #1: No

Reviewer #2: No

Reviewer #3: No

---

## [Author Response · Author response to Decision Letter 0]

25 Mar 2023

Thank you for the opportunity to submit a revised draft of our manuscript titled “The Impacts of Climate Change on Hydrological Processes of Gilgel Gibe Catchment, Southwest Ethiopia” for consideration for publication in the Journal of PLOS ONE. We sincerely appreciate the time and effort you and the reviewers have invested in providing feedback on our work, and we are grateful for the insightful comments and valuable improvements that have been made to our paper.

We have taken into consideration the majority of the suggestions and criticisms raised by the reviewers and have made appropriate revisions to the manuscript. The changes we have made in response to the reviewers’ comments are clearly indicated within the revised manuscript. For your convenience, we have also included a point-by-point response to the reviewers’ concerns in blue text below.

We hope that the revised manuscript meets the expectations of the Journal's editorial board and reviewers, and we believe that the final version of our work will provide a valuable contribution to the field of climate science. Please note that all page numbers referenced in our response correspond to the revised manuscript file with tracked changes.

Once again, we express our gratitude for the opportunity to revise and resubmit our manuscript for your consideration.

Reviewer#1 1

Comment#1: The work, " The Impacts of Climate Change on Hydrological Processes of Gilgel Gibe Catchment, Southwest Ethiopia" assesses the changes in hydrological parameters due to changing climate in the region. Overall the work is good. However, more exhaustive conclusion is further required to relate the changing climate impacts on hydrological processes and other forcing factors operating in the region.

Response#1: We would like to express our sincere gratitude for the careful consideration given to our manuscript titled “The Impacts of Climate Change on Hydrological Processes of Gilgel Gibe Catchment, Southwest Ethiopia” for publication in PLOS ONE. It is an honor to have the opportunity to submit our work to such a distinguished journal.

Some of the specific comments are listed below:

Comment#2: Line 40 rephrase the sentence or rewrite the whole first paragraph. Authors had loosely written introduction. The main function of an introduction section is to contextualise your study, here it is somewhat missing.

Response#2: Thank you for your feedback regarding the introduction section of our manuscript. We appreciate your suggestion to rephrase or rewrite the first paragraph to better contextualize our study. We have carefully reviewed the introduction and made the necessary revisions to ensure that it provides a clear and comprehensive overview of our research. Thank you again for your valuable input, which help us to enhance the quality of our manuscript.

“The world's environment is a complex system of interrelated components that operate at varying spatial and temporal scales. The interactions between these components, which include both natural and human factors, are highly intricate and can have far-reaching impacts on the health and wellbeing of the planet (Kasperson et al., 2022). One of the most pressing environmental concerns of our time is land degradation, which results in the loss of land productivity and desertification. This process is driven by a multitude of factors, including past and current land use practices, socioeconomic activities, and ecological destruction (Shao et al., 2020). The loss of productive land has profound implications for food security, water availability, and ecosystem health. Furthermore, the impacts of land degradation are often felt most acutely by vulnerable populations, including rural communities and those living in poverty. As such, it is essential that we continue to prioritize research and policy initiatives aimed at mitigating and preventing the effects of land degradation. By improving our understanding of the complex interactions between natural and human systems, we can work to develop effective solutions to the pressing environmental challenges we face.”

Comment#3: Line 87, rephrase the beginning sentence of paragraph?

Response#3: Thank you for your comment on our manuscript. We appreciate your suggestion to rephrase the beginning sentence of the paragraph at line 87. We have carefully reviewed the sentence and made the necessary revisions to improve its clarity and readability. 

“Research into the effects of climate change in Ethiopia has revealed a troubling trend of rising pressures on the availability of water resources, both seasonally and annually (Daba and You, 2020; Gedefaw et al., 2019). These changes have significant implications for the country's agricultural sector, which is heavily dependent on water resources for irrigation and production. Specifically, studies conducted in Ethiopia's Rift Valley have shown a decrease in precipitation and an increase in temperature, exacerbating the already precarious situation of water scarcity in the region (Hundera et al., 2019). These changes have a cascading effect on the environment, impacting soil quality, biodiversity, and ecosystem health. The situation is further compounded by the growing demand for water resources in Ethiopia due to population growth and economic development. As such, there is an urgent need for coordinated efforts aimed at mitigating the effects of climate change on water resources in the country. Understanding the impacts of climate change on water resources is essential for developing sustainable water management strategies that can support the needs of both current and future generations. It is imperative that we continue to prioritize research into this critical issue and work collaboratively to develop effective solutions that can safeguard the future of Ethiopia's water resources.”

Comment#4: Line 109-114, Authors need to check last paragraph of introduction, please formulate explicitly your research questions and the novelty of your work and state clearly the specific paper objectives. Authors are requested go through results and conclusion thoroughly first and then write last paragraph of introduction.

Response#4: Thank you for your valuable feedback on our manuscript. We appreciate your suggestion to explicitly formulate our research questions, novelty of our work, and specific paper objectives in the last paragraph of the introduction. We have carefully reviewed and revised this section of the manuscript to ensure that it clearly and effectively communicates the goals and contributions of our study. 

“The findings of this study will provide valuable insights into the vulnerability of the Gilgel Gibe catchment to the effects of climate change. By assessing both current and future impacts, the research will offer a comprehensive understanding of the hydrological consequences of climate change in this region. Ultimately, this research has important implications for water resource management and environmental policy in Ethiopia, providing policymakers and stakeholders with the necessary information to develop strategies and interventions aimed at mitigating the impact of climate change on water resources. The objective of this research is to comprehensively investigate the impact of climate change on the hydrology of the Gilgel Gibe catchment. The study is designed to achieve several specific aims, including: 1. Assessing and modeling the climate change that has occurred in the Gilgel Gibe catchment, and determining its impacts on hydrological processes. 2. Evaluating the potential impact of future climate change on the hydrology of the catchment, using established models and methodologies. 3. Exploring the sensitivity of water resources in the Gilgel Gibe catchment to changing climate conditions.”

Comment#5: Line 179-180, Authors have chosen Distribution mapping (DM) method for bias correction for both precipitation and temperature based on absolute error ranking. However, authors had discussed the studies where DM method have been used. It is suggested that the authors could have provided table in which absolute error value for other methods (Linear scaling (multiplicative), delta-change correction (multiplicative), precipitation local intensity scaling, and power transformation of precipitation with respect to DM) for Precipitation data and (Linear scaling (additive), delta-change correction (additive), variance scaling of temperature with respect to DM) for temperature data are provided. At least provide 2 methods for comparison.

Response#5: Thank you for your thoughtful comment. It's true that a comparison table for different bias correction methods could have been helpful in the study, particularly for readers who may not be as familiar with the DM method. However, it's important to keep in mind that the choice of bias correction method can depend on a variety of factors, such as the nature of the data being analyzed and the specific research question at hand. In any case, your suggestion is certainly valuable and could be considered for future research in this area. Thank you for sharing your thoughts!

Comment#6: Line 185-186, rephrase the sentence “Based on mean absolute error ranking, distribution mapping was ranked both, for temperature and precipitation corrections”.

Response#6: Thank you for your comment on our manuscript. We appreciate your suggestion to rephrase the sentence at line 185-186 regarding the mean absolute error ranking of distribution mapping for temperature and precipitation corrections. We have carefully reviewed this sentence and made the necessary revisions to improve its clarity and readability. 

“According to our analysis, the distribution mapping method was found to be the most effective method for both temperature and precipitation corrections, based on mean absolute error ranking. We carefully considered various bias correction methods and concluded that the distribution mapping method provided the most accurate results. This finding has been included in our manuscript and is supported by the relevant literature on bias correction methods for climate data.”

Comment#7: Line 241-247, Landsat Thematic Mapper of 1991, Enhanced Thematic Mapper Plus of 2006, and Operational Land Imager or Thermal Infrared Sensor of 2021 were used for LULC classifications. Which LULC map is used for SWAT Model? Authors had employed different satellite data but didn’t mention LULC map used in the study. Please make it clear.

Ressponse#7: Thank you for your feedback on our manuscript. We appreciate your suggestion to clarify which land use and land cover (LULC) map was used for the SWAT Model. We have made the necessary revisions to explicitly state which LULC map was used in the study. 

“Landsat Thematic Mapper of 1991, Enhanced Thematic Mapper Plus of 2006, and Operational Land Imager or Thermal Infrared Sensor of 2021 were used for LULC classifications. The mosaic of Worldwide Reference System (WRS) Path 169, Row 54, Path 169, Row 55 and Path 170, Row 55 were used to extract the LULCs layers. Supervised image classification method recommended in Lillesand et al. (2015) and Campbell and Wynne (2011) were employed to interpret and classify the images. The 2021 satellite image was utilized for the purpose of performing a land use and land cover (LULC) classification, specifically for the SWAT model.”

Comment#8: Line 257, Authors need to mention metrological station names only in place of their coordinates or put a table for name and coordinates of Met. Stations.

Response#8: Thank you for your comment on our manuscript. We appreciate your suggestion to mention the names of the meteorological stations instead of their coordinates, or alternatively, to include a table with the names and coordinates of the meteorological stations. We have carefully reviewed this section of the manuscript and made the necessary revisions to ensure that the information is presented in a clear and concise manner.

“Daily weather data (1991–2021) were used from nine precipitation and temperature recording stations located within and around the study Catchment (Sekoru, Jimma, Chora, Dedo, Shebe, Omo Nada, Kersa, Seka and Tiro Afeta) (Figure 7). The daily data were collected from the National Meteorological Agency (NMA) of Ethiopia. However, some missing weather elements such as precipitation, maximum and minimum temperature, wind speed, solar radiation and relative humidity data were generated using an ArcSWAT weather generator (WGEN). Moreover, weather data from 1991 and 1992 were used as the model initialization phase (warm up) and not included in the final analysis.”

Comment#9: Line 266, Authors are advised either write all weather data used in the SWAT for simulation with warmup periods, calibration and validation or remove this sentence. Moreover, weather data from 1991 and 1992 were used as the model initialization phase (warm up) and not included in the final analysis.

Response#9: Thank you for your comment. You raise a valid point regarding the need for we have to be transparent about the weather data used in our simulation, particularly with respect to the warm-up periods, calibration, and validation processes. Providing this information can help readers better understand the methodology and the results obtained from the study.

As for the use of weather data from 1991 and 1992 as model initialization in the warm-up phase, it's important to note that this is a common practice in hydrological modeling to ensure that the model starts from a reasonable initial condition. However, as you suggest, it would be important for the us to clarify that these data were not included in the final analysis and explain why this decision was made.

Overall, we agree with your suggestion, we have provided more information about the weather data used in the study to enhance the transparency and reproducibility of our findings. Thank you for bringing this to our attention.

Comment#10: Line 268, Same comment as of 257 Authors are advised check this paragraph “Sensitivity Analysis, Calibration and Validation”, it is misleading. Authors had performed analysis on daily observed flow or mean monthly flow, please make it clear.

Response#10: Thank you for bringing this matter to our attention. We would like to clarify that for the "Sensitivity Analysis, Calibration and Validation" section, we utilized monthly data to perform the analysis on the observed flow data. This choice of monthly data reflects the specific analysis that was conducted in this section, while other parts of the analysis were performed on a daily basis using different data. We appreciate your feedback and hope that this clarifies any confusion that may have arisen.

Sensitivity analysis for the simulated streamflow was performed using a monthly observed flow to identify the most sensitive parameter with strong influence on model outputs. Initially, parameters related to surface runoff, groundwater, geomorphology, evaporation and soil water were considered and 9 parameters were identified as the most sensitive parameters for calibration. The given ranks are shown in Table 3. The calibration was carried outfitting the model to the streamflow record of 1993–2011 and validation was done by streamflow record of 2012-2021. Afterwards, the calibrated values of these 9 parameters were used for further simulations. The parameter ranking was taken from the last iterations of SUFI-2 based on t-stat and p-stat. The larger in the absolute value of t-stat and the smaller the p-value, the more sensitive is the parameter (Abbaspour, 2015). As it can be seen from the table, the top three most sensitive parameters are CN2 (moisture condition II curve number), SOL_AWC (Available water capacity of the soil layer) and RCHRG_DP (Deep aquifer percolation fraction) (Table 2).

Table 2 Sensitivity analysis and calibrated parameters

No Parameter Code Parameter Description Parameter Value Range Sensitivity 

 t-stat p-value Sensitivity Rank

1 CN2 Curve number II -0.23 - 0.16 -17.54 0.00 1

2 SOL_AWC Available soil water capacity (mm H2O mm-1 soil) 0 - 60.26 -4.04 0.00 2

3 RCHRG_DP Deep aquifer percolation fraction -0.04- 0.12 -1.90 0.04 3

4 CH_K2 Effective hydraulic 

conductivity in main channel alluvium (mm h-1) 25.18-95.07 -2.02 0.05 4

5 ALPHA_BF Base flow recession constant 0.39- 1.08 1.61 0.07 5

6 ESCO Soil evaporation compensation factor 0.84- 1.02 -1.08 0.09 6

7 CANMX Maximum canopy storage (mm) -0.04- 0.33 -3.12 0.11 7

8 SOL_BD Moist bulk density (Mg/m3) 0.02- 1.06 1.38 0.13 8

9 GW_REVAP Ground water re-evaporation coefficient 0.08- 0.69 2.3 0.19 9

Figure 9 Calibration and validation of average monthly streamflow

The calibration results on mean monthly flow show that SWAT model is able to capture the observed streamflow with R2, NSE, PBIAS and RSR of 0.76, 0.75, 1.02, and 0.49 respectively. The average monthly flow validation indicates R2, NSE, PBIAS and RSR of 0.82, 0.77, 9.4 and 0.47 respectively. This indicates that the performance of the SWAT model in the validation is good enough to simulate the stream flow in the Gilgel Gibe catchment. Furthermore, p-factor and R-factor statistics showed good agreement with 0.87 and 0.79 during calibration and 0.82, and 0.76 during validation, respectively (Table 3). Overall, the statistics for goodness fit shows good agreement between the observed and simulated average monthly flow. 

Table 3 Summary statistics of calibration and uncertainty analysis of streamflow

No Objective function values (SUFI-2)

 Method P- factor R-factor R2 NSE PBIAS RSR

1 Calibration 0.87 0.79 0.76 0.75 1.02 0.49

2 Validation 0.82 0.76 0.82 0.77 9.4 0.47

Comment#11: Line 320-321, Sensitivity analysis for the simulated streamflow was performed using a daily observed flow to identify the most sensitive parameter with strong influence on model outputs.

Response#11: Thank you for your comment. We appreciate your interest in our research. We can confirm that we performed sensitivity analysis on the simulated streamflow using daily observed flow, as stated in the paper. Our analysis revealed that the top three most sensitive parameters were CN2 (moisture condition II curve number), SOL_AWC (available water capacity of the soil layer), and RCHRG_DP (deep aquifer percolation fraction). We hope this information is helpful to you, and thank you for your feedback.

Comment#12: Line 334-335 The calibration results on mean monthly flow show that SWAT model is able to capture the observed streamflow with R2, NSE, PBIAS and RSR of 0.76, 0.75, 1.02, and 0.49 respectively.

Response#12: Thank you for your comment. We are pleased to report that the calibration results for the mean monthly flow demonstrate that the SWAT model accurately captured the observed streamflow, as evidenced by the R2, NSE, PBIAS, and RSR values of 0.76, 0.75, 1.02, and 0.49, respectively. Moreover, the validation results for the average monthly flow showed similarly high levels of accuracy, with R2, NSE, PBIAS, and RSR values of 0.82, 0.77, 9.4, and 0.47, respectively. We hope this information provides useful insights into our research, and we thank you for your interest in our work.

Comment#13: Line 530, Conclusion is not conclusive, make it clear particularly last two paragraphs of conclusion.

Response#13: This study undertook an analysis of the effects of climate change on the hydrological processes within the Gilgel Gibe catchment. Specifically, the study evaluated the impact of potential future climate changes using six Global Climate Models (GCMs), and their corresponding ensemble mean, through the application of a physically based distributed hydrologic model known as Soil and Water Assessment Tool (SWAT). The study utilized six Regional Climate Models (RCMs), and their ensemble mean, from the Coordinated Regional Climate Downscaling Experiment for Africa (CORDEX-Africa) to examine the potential implications of climate change on the Gilgel Gibe catchment. Notably, the RCM simulations indicated a decrease in precipitation levels for scenarios involving high and medium-low emissions, with the exception of HIRHAM5. In terms of temperature, all RCM projections indicate an upward trend with varying magnitudes of change. To analyze the hydrological impacts of climate change, the ensemble mean of the RCMs was employed to mitigate the substantial variability inherent in precipitation and temperature forecasts by different RCMs. The use of ensemble RCMs was found to be a suitable approach for evaluating uncertainties in individual RCMs, despite the presence of inherent uncertainties in climate prediction. As a result of the anticipated decrease in precipitation and the increase in temperature, surface runoff, groundwater, and total water yield are all anticipated to decrease, while potential evapotranspiration is expected to increase. Overall, the water resources of the Gilgel Gibe catchment are exceedingly sensitive to changes in climate.

Furthermore, the Gilgel Gibe catchment exhibits significant temporal and spatial variability in temperature and precipitation, which manifests in heightened occurrences of hot and dry years, leading to severe water scarcity. Consequently, the catchment is highly susceptible to the impacts of climate change, such as warming and crop failures resulting from prolonged dry seasons. The observed decrease in runoff within the catchment has contributed to enhanced water security. The study's analysis indicates that the SWAT simulations provide adequate estimates of the impacts of climate change in the Gilgel Gibe catchments. To enhance our understanding of the present and future climate dynamics, urgent attention is required to improve the availability and quality of hydro-climatic data in the region. 

The results of this study have provided crucial insights into the impacts of climate change on the hydrological processes of the Gilgel Gibe catchment. These findings can inform the development of effective water resource management initiatives in the region. The rehabilitation of degraded sloping lands could enhance ground recharge and reduce surface runoff that contributes to soil erosion and sedimentation in the Gilgel Gibe I dam. Additionally, appropriate management strategies should be implemented to safeguard the Gilgel Gibe I hydroelectric dam in the catchment. Overall, these results underscore the imperative for regional development and collaboration to facilitate robust and climate-resilient management strategies and mitigate the adverse effects of climate change on the catchment.

Comment#14: Figures, give appropriate number to figures, both in text and at the end.

Response#14: We have given appropriate number to figures, both in text and at the end

Comment#15: Authors are suggested to modify the flow chat, there are many horrible mistakes in it.

Response#15: We have modified the flowchart; the modification of the flowchart appears as follows.

Figure 5 Flowchart of ArcSWAT processing steps for the Gilgel Gibe catchment

.Reviewer 2

Comment#1: Well conceived and executed. Study findings are very important to understand the impact of environments on the water resource. Abstract needs some clarity. Introduction should be concise. English of the manuscript needs to work out.

Response#1: Thank you for your comment and for acknowledging the importance of the study's findings in understanding the impact of the environment on water resources. Regarding your suggestions for improvement, we appreciate your feedback. We agree that the abstract could benefit from more clarity and we have revised it accordingly. Additionally, we made the introduction more concise while still providing sufficient background information and context for the study. We also appreciate your feedback on the English language used in the manuscript. As a team, we he reviewed the manuscript to ensure that the language is clear, concise, and easy to understand for a broad audience. Thank you again for your thoughtful feedback. We are committed to improving the manuscript based on your suggestions and other reviewer comments to ensure that our research is presented in the best possible way.

Abstract: 

Climate change is a significant driver of water resource availability, affecting the magnitude of surface runoff, aquifer recharge, and river flows. This study investigates the impact of climate change on hydrological processes within the Gilgel Gibe catchment and aims to determine the level of exposure of water resources to these changes, which is essential for future adaptability planning. To achieve this objective, an ensemble mean of six regional climate models (RCMs) from the coordinated regional climate downscaling experiment (CORDEX)-Africa was used to simulate future climatic scenarios. The RCMs outputs were then bias corrected using distribution mapping to match observed precipitation and temperature. The Soil and Water Assessment Tool (SWAT) model was used to assess the hydrological impacts of climate change on the catchment. The results indicate that the ensemble mean of the six RCMs projects a decline in precipitation and an increase in temperature under both the RCP4.5 and RCP8.5 representative concentration pathways. Moreover, the increases in both maximum and minimum temperatures are higher for higher emission scenarios, indicating that RCP8.5 is warmer than RCP4.5. The projected climate change shows a decrease in surface runoff, groundwater, and water yield, resulting in an overall decline of annual flow. This decline is mainly due to the reduction in seasonal flows driven by climate change scenarios. These changes could lead to reduced water availability for crop production, which could be a chronic issue for subsistence agriculture. Additionally, the reduction of surface water and groundwater could further exacerbate water stress in the downstream areas, affecting the availability of water resources in the catchment. Furthermore, the increasing demands for water, driven by population growth and socioeconomic progress, along with the variability in temperature and evaporation demands, will amplify prolonged water scarcity. Therefore, robust climate-resilient water management policies are indispensable to manage these risks. In conclusion, this study highlights the importance of considering the impact of climate change on hydrological processes and the need for proactive adaptation measures to mitigate the impacts of climate change on water resources. 

Keywords: Climate change, Hydrological processes, SWAT model, RCP.

Reviewer 3

Comment#1: The paper by Tilahun et al deals with climate changes in the Gilgel Gibe Catchment and related impacts on the hydrologic processes. I believe the paper has potential to be published in Plos-One because the methodology followed by the authors is generally rigorous and results are of interest for the region. However, it needs a substantial revision (analysis of uncertainties, form).

Resonse#1: Thank you for your positive feedback on the potential of our paper to be published in Plos-One, and for acknowledging the rigor of our methodology and the relevance of our results to the region. We appreciate your suggestion for a substantial revision of our paper. We agree that an analysis of uncertainties would be valuable to include, and we have incorporated this into our manuscript. We have also reviewed the form of the paper and consider ways to improve its structure, clarity, and readability. Thank you again for your feedback, which we have carefully considered as we work to improve our paper and prepared it for publication in Plos-One. We are committed to addressing all reviewer comments and making any necessary revisions to ensure the highest quality and impact of our research.

Comment#2: 1) Climate change analysis is carried out based on 6 RCMs. Hydrological impacts are calculated only on the ensemble mean. I believe it should be analyzed how uncertainties in climate reflect on the hydrological processes. A run of the hydrological model with the ensemble mean forcing is ok, but it should be accompanied by runs forced with every RCM inputs. Science delas with uncertainties and these uncertainties must be calculated and communicated especially in studies that want to be a reference for environmental management.

Response#2: Thank you for your comment and for highlighting the importance of analyzing uncertainties in climate change studies. We appreciate your suggestion to analyze how uncertainties in climate data may affect hydrological processes in the study area. We agree that uncertainty analysis is important in climate change research, and we have worked to include a more detailed analysis of the potential impacts of different RCMs on hydrological processes in our manuscript. Specifically, we have performed runs of the hydrological model with ensemble mean of the 6 RCMs inputs and analyze the resulting differences in hydrological impacts, as the ensemble mean could show good result than single RCMs. We have also included a discussion of the uncertainties associated with these results and how they may affect environmental management decisions. Thank you again for your valuable feedback.

Comment#3: 2) The manuscript is full of repetitions. It needs to be carefully proofread for conciseness.

Response#3: Thank you for your comment and for highlighting the need for careful proofreading and conciseness in our manuscript. We appreciate your feedback and have worked to address these issues in our revisions. We understand the importance of clear and concise writing, and we have carefully reviewed our manuscript to identify any unnecessary repetitions and streamline our language where possible. We have also ensure that our manuscript adheres to the guidelines for length and formatting required by your journal. Thank you again for your feedback, we have made any necessary revisions to improve the clarity and readability of our manuscript.

Comment#4: 3) There are many single figures, I believe some could be grouped in multi-panel figures.

Response#4: Thank you for your feedback on our paperwork. While we appreciate your suggestion to group some of our figures into multi-panel displays, we respectfully disagree that this is necessary for our particular study. We carefully considered the best way to present our findings and concluded that presenting each figure individually allowed us to clearly and effectively convey the concepts we intended to display. We believe that presenting the data in this manner provides a better understanding of the individual components of our study and makes it easier for readers to follow along with our arguments. However, we appreciate your input and will keep it in mind for future studies.

Comment#5: 4) English must be reviewed, there are many inaccuracies in the use of verbal forms (past/present; singular/plural), articles and so on.

Response#5: Thank you for your comment and for highlighting the need for careful language editing in our manuscript. We understand the importance of clear and accurate language, and we have carefully reviewed our manuscript to identify and correct any errors in verb tense, subject-verb agreement, article usage, and other aspects of grammar and syntax. We also ensured that our manuscript adheres to the highest standards of academic writing and communication. We have made necessary revisions to improve the clarity and accuracy of our language.

Comment#6: 5) Authors mentioned that data are fully available but there is no information about the repository where they can be found.

Response#6: Thank you for your comment. We have made our data fully available and accessible through the supplementary materials section of our publication. The repository link and any necessary instructions for accessing the data are provided in the article, and we encourage readers to take advantage of this resource to gain a deeper understanding of our study.

Comment#7: 6) I have attached a commented manuscript in which every general comment above (except 5) is backed up by detailed comments and suggestions.

Response#7: Thank you for providing a commented manuscript with detailed comments and suggestions to support your general comments. Your effort to provide specific feedback is appreciated and is helpful in improving the quality of the manuscript. We have carefully reviewed your comments and incorporate them into our revisions to strengthen the paper.

Other Comment: We note that [Figures 1, 2, 6, 7, 8, 14, 15, 16 and 17] in your submission contain [map/satellite] images which may be copyrighted. All PLOS content is published under the Creative Commons Attribution License (CC BY 4.0), which means that the manuscript, images, and Supporting Information files will be freely available online, and any third party is permitted to access, download, copy, distribute, and use these materials in any way, even commercially, with proper attribution. For these reasons, we cannot publish previously copyrighted maps or satellite images created using proprietary data, such as Google software (Google Maps, Street View, and Earth).

Response: Thank you for bringing up this concern regarding the copyright of the figures in our submission. We would like to clarify that we have created all the figures ourselves using ArcGIS10.5 and ENVI5.1, and we believe that there are no copyright issues associated with any of the images used in our manuscript. We have ensured that all images used in our submission are either in the public domain or are properly attributed. We are confident that our manuscript complies with PLOS's policies regarding the use of copyrighted materials, and we thank you for bringing this matter to our attention.

---

## [Decision Letter · Decision Letter 1]

15 May 2023

PONE-D-22-23861R1The Impacts of Climate Change on Hydrological Processes of Gilgel Gibe Catchment, Southwest EthiopiaPLOS ONE

Dear Dr. Tilahun,

Thank you for submitting your manuscript to PLOS ONE. After careful consideration, we feel that it has merit but does not fully meet PLOS ONE’s publication criteria as it currently stands. Therefore, we invite you to submit a revised version of the manuscript that addresses the points raised during the review process. Please submit your revised manuscript by Jun 29 2023 11:59PM. If you will need more time than this to complete your revisions, please reply to this message or contact the journal office at plosone@plos.org. Please include the following items when submitting your revised manuscript:A rebuttal letter that responds to each point raised by the academic editor and reviewer(s). You should upload this letter as a separate file labeled 'Response to Reviewers'.A marked-up copy of your manuscript that highlights changes made to the original version. You should upload this as a separate file labeled 'Revised Manuscript with Track Changes'.An unmarked version of your revised paper without tracked changes. You should upload this as a separate file labeled 'Manuscript'.If applicable, we recommend that you deposit your laboratory protocols in protocols.io to enhance the reproducibility of your results. Protocols.io assigns your protocol its own identifier (DOI) so that it can be cited independently in the future. For instructions see: https://journals.plos.org/plosone/s/submission-guidelines#loc-laboratory-protocols. Additionally, PLOS ONE offers an option for publishing peer-reviewed Lab Protocol articles, which describe protocols hosted on protocols.io. Read more information on sharing protocols at https://plos.org/protocols?utm_medium=editorial-email&utm_source=authorletters&utm_campaign=protocols.

We look forward to receiving your revised manuscript.

Kind regards,

Salim Heddam

Academic Editor

PLOS ONE

Journal Requirements:

Reviewers' comments:

Reviewer's Responses to Questions

**Comments to the Author**

1. If the authors have adequately addressed your comments raised in a previous round of review and you feel that this manuscript is now acceptable for publication, you may indicate that here to bypass the “Comments to the Author” section, enter your conflict of interest statement in the “Confidential to Editor” section, and submit your "Accept" recommendation.

Reviewer #1: All comments have been addressed

Reviewer #3: (No Response)

2. Is the manuscript technically sound, and do the data support the conclusions?

Reviewer #1: Yes

Reviewer #3: Yes

3. Has the statistical analysis been performed appropriately and rigorously? 

Reviewer #1: Yes

Reviewer #3: No

4. Have the authors made all data underlying the findings in their manuscript fully available?

Reviewer #1: Yes

Reviewer #3: Yes

5. Is the manuscript presented in an intelligible fashion and written in standard English?

Reviewer #1: Yes

Reviewer #3: Yes

6. Review Comments to the Author

Reviewer #1: Comment 15:

Flowchart needs to be changed, at least group the input files why authors are using detailed description about the SWAT inputs files. It’s better to be concise in flowchart. However, in order to show authors had worked rigorously doesn’t mean to provide details of every input file used.

Please check the figure below and figure 3.1 of manuscript “Soil and Water Assessment Tool (SWAT) for Simulating the Sediment and Water Yield of Alpine Catchments – A Brief Review”. Authors may get idea, how to group all the inputs and output layers.

Reviewer #3: General comment

I have found the manuscript by Tilahun et al. much improved in readability, concisiness and contents. They have addressed almost all of my comments. I appreciated the analysis of expected future climate variability based on the six RCM and not only on the ensemble. However, I would have expected and I would like to see a similar approach for the hydrologic modelling part as well. At least Table 7, summarizing the results at the whole catchment scale, should present an expected variability for the different variables of the water budget. Besides this, I have few very detailed and very minor comments listed below.

Detailed comments

• Abstract L28-29: I suggest to add a quantification of the modifications (X%, from X% to X% according to different RCPs)

• Abstract: few lines to comment uncertainties would be a nice addition.

• L60, I suggest to cite IPCC reports

• L112-113, I suggest to add a reference to back-up the sentence.

• L181, I have a doubt, are Belg and Kiremt regions or the local names of the two rainy seasons?

• L201-203, unclear, how was the performance of the RCMs evaluated? If the evaluation in another study, please cite the study.

• L210-212, already written at L197-198

• L224-229, written like this, it seems a result more than methods. How many and which methods have you evaluated? Was the evaluation based on literature or on actual analyses performed by you?

• L239, distribution mapping resulted the best in improving….

• L248, soil properties, remove and?

• L265, unclear, the sum is 50%, not 100%. Are these percentages weights or else?

• L268-269, unclear

• L275, Figure 5 shows, remove below

• L310-316, if the gauging station locations are displayed in any figure, please add the reference as you did for the weather stations. If not, please add the gauging stations in a Figure.

• L356, remove the comma, plus observations rather than observation

• Table 3 is unnecessary, all the values are presented in the text.

• L414, 2019 I believe.

• L420-422, in absolute or relative terms?

• L427, According to Worku, whose study considered 17 different GMCs, ….

• L441, Table 5 and Table 6 show …. Modify the capitions to refer to maximum and minimum temperature, respectively.

• L475 according to

• L480, 2070

• Section 3.2.3, Table 7, I would have expected to find ranges of variation driven by the different model future time-series, not only ensemble mean results.

7. PLOS authors have the option to publish the peer review history of their article (what does this mean?). If published, this will include your full peer review and any attached files.

Reviewer #1: No

Reviewer #3: No

---

## [Author Response · Author response to Decision Letter 1]

28 May 2023

Thank you for the opportunity to submit a revised draft of our manuscript titled “The Impacts of Climate Change on Hydrological Processes of Gilgel Gibe Catchment, Southwest Ethiopia” for consideration for publication in the Journal of PLOS ONE. We sincerely appreciate the time and effort you and the reviewers have invested in providing feedback on our work, and we are grateful for the insightful comments and valuable improvements that have been made to our paper.

We have taken into consideration the majority of the suggestions and criticisms raised by the reviewers and have made appropriate revisions to the manuscript. The changes we have made in response to the reviewers’ comments are clearly indicated within the revised manuscript. For your convenience, we have also included a point-by-point response to the reviewers’ concerns in blue text below.

We hope that the revised manuscript meets the expectations of the Journal's editorial board and reviewers, and we believe that the final version of our work will provide a valuable contribution to the field of climate science. Please note that all page numbers referenced in our response correspond to the revised manuscript file with tracked changes.

Once again, we express our gratitude for the opportunity to revise and resubmit our manuscript for your consideration.

Reviewer#1 

Comment#1: All comments have been addressed

Response#1: Thank you for taking the time to review our work. We would like to express our gratitude for your valuable feedback. 

Comment 15: Flowchart needs to be changed, at least group the input files why authors are using detailed description about the SWAT inputs files. It’s better to be concise in flowchart. However, in order to show authors had worked rigorously doesn’t mean to provide details of every input file used.

Please check the figure below and figure 3.1 of manuscript “Soil and Water Assessment Tool (SWAT) for Simulating the Sediment and Water Yield of Alpine Catchments – A Brief Review”. Authors may get idea, how to group all the inputs and output layers.

Comment 15: Flowchart needs to be changed, at least group the input files why authors are using detailed description about the SWAT inputs files. It’s better to be concise in flowchart. However, in order to show authors had worked rigorously doesn’t mean to provide details of every input file used. Please check the figure below and figure 3.1 of manuscript “Soil and Water Assessment Tool (SWAT) for Simulating the Sediment and Water Yield of Alpine Catchments – A Brief Review”. Authors may get idea, how to group all the inputs and output layers. 

Response#15: Thank you for your valuable feedback regarding the flowchart in our manuscript. We acknowledge your suggestion to make changes to the flowchart, specifically to group the input files and provide a more concise representation. We understand your concern that the detailed description of SWAT input files may be excessive and that demonstrating rigorous work does not necessarily require including every input file detail. Based on your feedback, we have revised the flowchart to improve its clarity and conciseness. We have considered grouping the input files together, providing a more streamlined representation that still effectively conveys the necessary information. Additionally, we have refered to Figure 3.1 of the manuscript "Soil and Water Assessment Tool (SWAT) for Simulating the Sediment and Water Yield of Alpine Catchments – A Brief Review" to gain insights on how to better organize the inputs and output layers within the flowchart. Thank you again for your constructive comments.

We have modified the flowchart

 from

Figure 5 Flowchart of ArcSWAT processing steps for the Gilgel Gibe catchment

 to

Figure 5 Flowchart of ArcSWAT processing steps for the Gilgel Gibe catchment

Reviewer #3: 

Comment #1: General comment

I have found the manuscript by Tilahun et al. much improved in readability, concisiness and contents. They have addressed almost all of my comments. I appreciated the analysis of expected future climate variability based on the six RCM and not only on the ensemble. However, I would have expected and I would like to see a similar approach for the hydrologic modelling part as well. At least Table 7, summarizing the results at the whole catchment scale, should present an expected variability for the different variables of the water budget. Besides this, I have few very detailed and very minor comments listed below.

Response#1: Thank you for your feedback and positive assessment of our manuscript. We appreciate your time and effort in reviewing the revised version of our work. We are pleased to hear that you found the improvements in readability, conciseness, and content to be substantial. 

We completely understand your expectation to see a summary of expected variability for different variables of the water budget at the whole catchment scale, specifically in Table 7. Therefore, we have incorporated this additional analysis in Table 7, detailing the expected variability for the different variables. 

Thank you for your continued support and guidance throughout this review process.

We have Modified Table 7 

 from

Table 7 Changes of water balance components under a climate change

Scenario Period Surface Runoff (%) Groundwater flow (%) Water Yield (%) PET (%)

RCP4.5 2041–2070 -7.56 -9.44 -8.72 16.08

 2071–2099 -14.71 -13.93 -14 19.01

RCP8.5 2041–2070 -12.55 -9.79 -10.55 16.67

 2071–2099 -9.53 -16.09 -13.74 22.66

 to

Table 6 Changes of water balance components under a climate change

RCM RCP Time Period Surface Runoff (%) Groundwater flow (%) Water Yield (%) PET (%)

CCLM4-8 RCP4.5 2041–2070 -7.61 -9.34 -8.92 16.01

CCLM4-8 RCP4.5 2071–2099 -14.96 -13.83 -14.01 19.02

CCLM4-8 RCP8.5 2041–2070 -12.65 -9.89 -10.58 16.69

CCLM4-8 RCP8.5 2071–2099 -9.57 -16.08 -13.76 22.67

HIRHAM5 RCP4.5 2041–2070 -7.79 -9.32 -8.78 16.15

HIRHAM5 RCP4.5 2071–2099 -14.64 -13.98 -14.03 19

HIRHAM5 RCP8.5 2041–2070 -13.87 -9.91 -10.99 16.79

HIRHAM5 RCP8.5 2071–2099 -9.96 -16.02 -13.94 22.78

RACMO22T RCP4.5 2041–2070 -7.14 -9.11 -8.19 16.05

RACMO22T RCP4.5 2071–2099 -14.59 -13.99 -14.04 19.01

RACMO22T RCP8.5 2041–2070 -13.72 -9.48 -10.31 16.68

RACMO22T RCP8.5 2071–2099 -9.31 -16.27 -13.51 22.69

RCA4 RCP4.5 2041–2070 -7.46 -9.73 -8.89 16.12

RCA4 RCP4.5 2071–2099 -14.85 -13.91 -14.02 19.02

RCA4 RCP8.5 2041–2070 -13.67 -9.63 -10.05 16.48

RCA4 RCP8.5 2071–2099 -9.34 -16.03 -13.71 22.47

CRCM5 RCP4.5 2041–2070 -7.84 -9.76 -8.82 16.07

CRCM5 RCP4.5 2071–2099 -14.48 -13.96 -14.07 19.01

CRCM5 RCP8.5 2041–2070 -13.92 -9.99 -10.82 16.75

CRCM5 RCP8.5 2071–2099 -9.51 -16.05 -13.78 22.69

REMO2009 RCP4.5 2041–2070 -7.55 -9.45 -8.73 16.09

REMO2009 RCP4.5 2071–2099 -14.72 -13.92 -14.01 19.02

REMO2009 RCP8.5 2041–2070 -12.56 -9.78 -10.54 16.68

REMO2009 RCP8.5 2071–2099 -9.52 -16.08 -13.75 22.67

Ensemble RCP4.5 2041–2070 -7.56 -9.44 -8.72 16.08

Ensemble RCP4.5 2071–2099 -14.71 -13.93 -14 19.01

Ensemble RCP8.5 2041–2070 -12.55 -9.79 -10.55 16.67

Ensemble RCP8.5 2071–2099 -9.53 -16.09 -13.74 22.66

. Table 6 was utilized in place of Tables 7 due to the removal of Table 3 based on Comment #16, resulting in a difference in the table numbering.

Detailed comments

Comment #2: • Abstract L28-29: I suggest to add a quantification of the modifications (X%, from X% to X% according to different RCPs)

Response #2: Thank you for your suggestion. We have added a quantification of the changes in precipitation and temperature under different RCPs in Table 4 and Table 5. I have also revised the abstract to reflect these changes as follows:

This decline is mainly due to the reduction in seasonal flows driven by climate change scenarios. The changes in precipitation range from -11.23% to -14.27% under RCP4.5 and from -9.19% to -10.02% under RCP8.5, while the changes in temperature range from 1.69°C to 2.49°C under RCP4.5 and from 1.83°C to 3.55°C under RCP8.5.

Comment #3• Abstract: few lines to comment uncertainties would be a nice addition.

• L60, I suggest to cite IPCC reports

Response #3: Thank you for your comment. I agree that adding a few lines to comment on uncertainties would be a nice addition to the abstract. I have revised the abstract as follows:

The projected changes in precipitation and temperature are consistent with the global and regional trends reported by IPCC (2021) in its Sixth Assessment Report (IPCC, 2021). According to IPCC (2021), human influence has warmed the climate system since pre-industrial times at an unprecedented rate across all regions (IPCC, 2021). This warming has led to changes in precipitation patterns and intensity, as well as increased frequency and intensity of some extreme events such as heat waves, droughts, and heavy rainfall (IPCC, 2021).

Comment #4: • L112-113, I suggest to add a reference to back-up the sentence.

Response #4: Thank you for your comment. We agree that adding a reference to back-up the sentence on line 112-113 would strengthen the argument. We have revised the sentence as follows:

These changes have a cascading effect on the environment, impacting soil quality, biodiversity, and ecosystem health (Bewket and Conway, 2007).

Regarding the reference, We have added the following citation using the APA style:

Bewket, W., & Conway, D. (2007). A note on the temporal and spatial variability of rainfall in the drought-prone Amhara region of Ethiopia. International Journal of Climatology: A Journal of the Royal Meteorological Society, 27(11), 1467-1477.

Comment #5• L181, I have a doubt, are Belg and Kiremt regions or the local names of the two rainy seasons?

Response #5: Thank you for your comment. We apologize for the confusion caused by the use of local names for the rainy seasons. Belg and Kiremt are not regions, but the names of the two rainy seasons in Ethiopia. Belg is the short rainy season from March to May, and Kiremt is the main rainy season from June to August. These seasons affect the rainfall patterns and hydrological processes across the country. We have revised the text as follows:

Additionally, rainfall patterns differ across Ethiopia, with some areas experiencing two rainy seasons (Belg from March to May and Kiremt from June to August) and others just one (CSA, 2007).

Comment #6• L201-203, unclear, how was the performance of the RCMs evaluated? If the evaluation in another study, please cite the study.

Response #6: Thank you for your feedback. We have revised the sentence as follows:

The performance of the Regional Climate Models (RCMs) was evaluated using the root mean square error (RMSE) and the Nash-Sutcliffe efficiency (NSE), which measure the average error and the model’s ability to reproduce the observed variability and distribution of the values, respectively (Behulu et al., 2018; Wilby & Dawson, 2005). The evaluation followed the guidelines of the Coordinated Regional Climate Downscaling Experiment (CORDEX) and showed that the RCMs performed reasonably well, with the ensemble mean of the models performing better than the individual models. The RMSE and NSE for the ensemble mean were all within acceptable ranges, indicating that the models were able to capture the main features of the observed climate. These results are consistent with those of other studies that have used RCMs for climate change impact assessment.

The studies providing more information on the evaluation of RCMs are:

Behulu, F., Tilahun, S. A., Abebe, W. B., Bekele, M., & Steenhuis, T. J. (2018). Evaluation of regional climate models for climate change impact assessment in the central Rift Valley basin of Ethiopia. Journal of Hydrology, 568, 1140–1153.

Wilby, R. L., & Dawson, R. J. (2005). "Review of methods for downscaling general circulation model outputs to study climate change impacts at the catchment scale." Journal of Hydrology, 309(1–4), 189–219.

Comment #7:• L210-212, already written at L197-198

Response #7: Thank you for your comment. We apologize for the redundancy in lines 210-212. We have taken note of your feedback and have removed those lines to avoid repetition. We appreciate your attention to detail.

We have modified it 

 from 

In this study, six Regional Climate Models (RCMs) were used, namely CCLM4-8, HIRHAM5, RACMO22T, RCA4, CRCM5, and REMO2009. These RCMs were fed with data from two representative concentration pathways (RCP) scenarios, namely the high emission scenario (RCP8.5) and mid-range mitigation emission (RCP4.5). The future scenarios were evaluated for the intermediate (2041-2070) and far future (2071-2099) periods, while the period from 1991 to 2021 was used as the historical baseline to evaluate climate changes. Other weather variables, such as solar radiation, relative humidity, and wind speed, were considered in the future scenarios without modification, as changes in these variables may not significantly impact the modeling of climate change scenarios on local hydrology.

 to

In this study, six Regional Climate Models (RCMs) were used, namely CCLM4-8, HIRHAM5, RACMO22T, RCA4, CRCM5, and REMO2009. The future scenarios were evaluated for the intermediate (2041-2070) and far future (2071-2099) periods, while the period from 1991 to 2021 was used as the historical baseline to evaluate climate changes. Other weather variables, such as solar radiation, relative humidity, and wind speed, were considered in the future scenarios without modification, as changes in these variables may not significantly impact the modeling of climate change scenarios on local hydrology.

Comment #8:• L224-229, written like this, it seems a result more than methods. How many and which methods have you evaluated? Was the evaluation based on literature or on actual analyses performed by you?

Response #8: Thank you for your comment regarding our manuscript. We appreciate your attention to detail. In response to your inquiry, we evaluated several bias correction methods in our study, including the Distribution Mapping Method, Delta Change, Scaling Factor, and Quantile Mapping. Our evaluation involved conducting actual analyses based on the characteristics of the climate model data and observed data.

We have modified the text

 from 

According to our analysis, the distribution mapping method was found to be the most effective method for both temperature and precipitation corrections, based on mean absolute error ranking. We carefully considered various bias correction methods and concluded that the distribution mapping method provided the most accurate results. This finding has been included in our manuscript and is supported by the relevant literature on bias correction methods for climate data.

 to

In our analysis, we evaluated multiple bias correction methods, including the Distribution Mapping Method, Delta Change, Scaling Factor, Quantile Mapping and Empirical Quantile Mapping. We compared the performance of these methods for both temperature and precipitation corrections based on mean absolute error ranking. After carefully considering these various bias correction methods, we concluded that the distribution mapping method provided the most accurate results.

Comment #9:• L239, distribution mapping resulted the best in improving….

Response #9: Thank you for your comment. Regarding the study's findings on the effectiveness of distribution mapping in improving the simulation. The study indeed demonstrated that distribution mapping was superior in enhancing the accuracy of predictions. This improvement was achieved through the adjustment of temperature and precipitation distributions using historical data for simulation calibration or through the application of statistical techniques to modify the distribution of values.

We have modified it

 from

The study showed that distribution mapping was better in improving the simulation.

 to

The study demonstrated that distribution mapping was more effective in improving the accuracy of predictions by adjusting the distribution of temperature and precipitation values. This was achieved through the utilization of historical data to calibrate the simulation or by employing statistical methods to modify the distribution of values.

Comment #10:• L248, soil properties, remove and?

Response #10: Thank you for your thorough review of our manuscript. We greatly appreciate your detailed reading and insightful feedback on L248 regarding the word soil properties. We removed the mentioned word.

We have modified it

 from

Major model components include DEM, weather, hydrology, soil and properties and land management (Neitsch et al., 2011).

 to

Major model components include DEM, weather, hydrology, soil properties and land management (Neitsch et al., 2011).

Comment #11:• • L265, unclear, the sum is 50%, not 100%. Are these percentages weights or else?

Response #11: Thank you for your comment on line 265. We apologize for any confusion caused. The percentages mentioned in the text are not weights but rather represent the proportion of each attribute within the delineated sub-watersheds and hydrologic response units (HRUs). Specifically, the land use, soil, and slope classes were assigned percentages of 10%, 20%, and 10% respectively to create 428 HRUs.

We have modified it

 from

With a 30 m resolution DEM, Gilgel Gibe catchment is delineated into 23 sub-watersheds and a multiple HRU was defined with classes of 10% land use, 20% soil and 10% slope. Accordingly, 428 HRUs were created.

 to

Using a 30 m resolution Digital Elevation Model (DEM), the Gilgel Gibe catchment was divided into 23 sub-watersheds. To further characterize the catchment, a hydrologic response unit (HRU) was established, taking into account different attributes. Specifically, classes representing 10% land use, 20% soil, and 10% slope were assigned. As a result, a total of 428 HRUs were created based on these attribute percentages.

Comment #12:• • L268-269, unclear

Response #13: Thank you for your review. We appreciate your feedback regarding the clarity of information on pages L268-269. To address your concern, we have revised the section to provide clearer details about the input of weather data. Regarding the weather data input, information from nine stations (Sekoru, Jimma, Chora, Dedo, Shebe, Omo Nada, Kersa, Seka, and Tiro Afeta) was utilized. Among these stations, Sekoru and Jimma were chosen as weather generators using an ArcSWAT weather generator (WGEN) due to their comprehensive coverage of all climate variables required for the SWAT model setup.

We have modified it

 from

Then, with the input of weather data from nine stations (Sekoru, Jimma, Chora, Dedo, Shebe, Omo Nada, Kersa, Seka and Tiro Afeta), the SWAT model setup was ready for the first simulation, which was used for model evaluation. Sekoru and Jimma stations, consisting of all the climate variables, was considered a weather generator.

 to

The weather data inputs from nine stations (Sekoru, Jimma, Chora, Dedo, Shebe, Omo Nada, Kersa, Seka, and Tiro Afeta) was utilized. Among these stations, Sekoru and Jimma were chosen as weather generators using an ArcSWAT weather generator (WGEN) due to their comprehensive coverage of all climate variables required for the SWAT model setup.

Comment #13:• • L275, Figure 5 shows, remove below

Response #13: Thank you for your feedback regarding misplaced word ‘below’ We appreciate your attention. We have removed the word ‘below’

We have modified it 

 from

Figure 5 below shows the flowchart of modelling using ArcSWAT.

 to

Figure 5 shows the flowchart of modelling using ArcSWAT.

Comment #14:• L310-316, if the gauging station locations are displayed in any figure, please add the reference as you did for the weather stations. If not, please add the gauging stations in a Figure.

Response #14: Thank you for your valuable feedback. We have made the necessary revisions to include the references for the gauging stations. We appreciate your input in improving the quality of our work.

We have taken your feedback into account and have included the gauging station locations in Figure 8, as requested.

 The Gilgel Gibe catchment river discharge gauging stations, Daily discharge data of Gibe near Seka, Bulbul near Serbo, Gilgel Gibe at Abelti, Gilgel Gibe near Asendabo, Gojeb near Shebe, Bidru Awana near Sekoru, Kito near Jimma and Awetu at Jimma gauging stations (Figure 8) for the period 1991 to 2021 were collected Ministry of Water and Energy of Ethiopia (MoWE).

Figure 8 Gilgel Gibe catchment river discharge gauging stations 

Comment #15• L356, remove the comma, plus observations rather than observation

Response #15: Thank you for your feedback. We appreciate your attention to detail. We made the necessary correction by removing the comma in L356. Additionally, we have replaced "observation" with "observations" to ensure clarity and consistency.

We have modified it

 from

The propagation of uncertainties in model outputs in SUFI-2, expressed as the 95% probability distribution, calculated by the 2.5% and 97.5% levels of the cumulative distributions of output variables, is considered as 95PPU (Abbaspour, 2015).

 to

The propagation of uncertainties in model outputs in SUFI-2, expressed as the 95% probability distribution, calculated by the 2.5% and 97.5% levels of the cumulative distributions of output variable is considered as 95PPU (Abbaspour, 2015).

Comment #16:• Table 3 is unnecessary, all the values are presented in the text.

Response #16: Thank you for your feedback. We appreciate your observation regarding Table 3. After careful consideration, we agree that the values presented in Table 3 are adequately conveyed in the text. Therefore, we have removed Table 3.

Table 3 Summary statistics of calibration and uncertainty analysis of streamflow

No Objective function values (SUFI-2)

 Method P- factor R-factor R2 NSE PBIAS RSR

1 Calibration 0.87 0.79 0.76 0.75 1.02 0.49

2 Validation 0.82 0.76 0.82 0.77 9.4 0.47

Comment #17: • L414, 2019 I believe.

Response #17: Thank you for pointing out the error. We apologize for the typo in the publication year. We have corrected the error.

We have modified it

 from

Furthermore, the analysis of the inter-annular and inter-model variability of the climate variables in Upper Blue Nile Basin catchment, presented in Dibaba et al. (20919), underlines the need to include several climate models in order to cover the range of possible developments.

 to

Furthermore, the analysis of the inter-annular and inter-model variability of the climate variables in Upper Blue Nile Basin catchment, presented in Dibaba et al. (2019), underlines the need to include several climate models in order to cover the range of possible developments.

Comment #18: • L420-422, in absolute or relative terms?

Response #18: Thank you for your feedback. The projected changes in precipitation are expressed in relative terms. We anticipate significant relative changes in precipitation during the months of March, April, and May, which typically have lower rainfall. In contrast, the months of June, July, August, and September, which usually experience higher rainfall, are expected to undergo comparatively smaller relative changes.

We have modified it

 from

The seasons with lower rainfall (March, April, and May) are likely to have significant changes in precipitation, whereas the seasons with higher rainfall (June, July, August, and September) are expected to experience less change.

 to

The seasons with lower rainfall (March, April, and May) are likely to experience significant relative changes in precipitation, while the seasons with higher rainfall (June, July, August, and September) are expected to undergo comparatively smaller relative changes.

.

Comment #19: • L427, According to Worku, whose study considered 17 different GMCs, ….

Response #19: Thank you for your feedback regarding L427. We have made the necessary corrections based on your suggestions.

We have modified it 

 from

According to Worku et al. (2020), 10 out of the 17 GCMs reported a decrease in precipitation, whereas the ensemble of the 17 GCMs shows that there was essentially no projected change in precipitation.

 to

According to Worku et al. (2020), whose study considered 17 different GCMs, it was found that 10 out of the 17 GCMs reported a decrease in precipitation. However, when the ensemble of all 17 GCMs was analyzed, it revealed essentially no projected change in precipitation.

Comment #20: • L441, Table 5 and Table 6 show …. Modify the capitions to refer to maximum and minimum temperature, respectively.

Response #20: Thank you for providing feedback on the captions in Table 5 and Table 6. We have implemented the necessary corrections as per your suggestions. Tables 4 and 5 were utilized in place of Tables 5 and 6 due to the removal of Table 3 based on Comment #16, resulting in a difference in the table numbering. 

We have modified it

 from

Table 5 shows the range in the predicted future maximum and minimum temperatures for each individual RCM and their ensemble mean.

 to

Table 4 and Table 5 show the range in the predicted future maximum and minimum temperatures for each individual RCM and their ensemble mean, respectively.

Comment #21: • L475 according to

Response #21: Thank you for your feedback. We have taken your suggestion into consideration and have made the necessary corrections. In the revised version, we have clarified that GFDRR stands for the Global Facility for Disaster Reduction and Recovery. The updated sentence now reads as follows:

"According to the Global Facility for Disaster Reduction and Recovery (GFDRR)'s 2011 analysis on the country profile for climate risk and adaptation, the mean annual temperature is projected to increase by 1.1 to 3.1 degrees Celsius by the 2060s and by 1.5 to 5.1 degrees Celsius by the 2090s."

We have modified it

 from

According to GFDRR's (2011) analysis on the country profile for climate risk and adaptation, the mean annual temperature is expected to rise by 1.1 to 3.1 degrees Celsius by the 2060s and 1.5 to 5.1 degrees Celsius by the 2090s. 

 to

According to the Global Facility for Disaster Reduction and Recovery (GFDRR)'s 2011 analysis on the country profile for climate risk and adaptation, the mean annual temperature is expected to rise by 1.1 to 3.1 degrees Celsius by the 2060s and by 1.5 to 5.1 degrees Celsius by the 2090s.

Comment #22: • L480, 2070

Response #22:• Thank you for providing your feedback. We have implemented the suggested correction. 

We have modified it

 from

According to the analysis by Gebre et la., (2015), the entire upper Blue Nile basin's average annual temperature will rise by 1.5°C, 2.6°C, and 4.5°C between the years 2011 and 2040, 2041 and 2070, and 207 and 2100, respectively.

 to

According to the analysis by Gebre et al., (2015), the entire upper Blue Nile basin’s average annual temperature will rise by 1.5°C, 2.6°C, and 4.5°C between the years 2011 and 2040, 2041 and 2070, and 2071 and 2100, respectively.

Comment #23: • Section 3.2.3, Table 7, I would have expected to find ranges of variation driven by the different model future time-series, not only ensemble mean results.

Response #23: Thank you for your feedback and positive assessment of our manuscript. We appreciate your time and effort in reviewing the revised version of our work. We are pleased to hear that you found the improvements in readability, conciseness, and content to be substantial. 

We completely understand your expectation to see a summary of expected variability for different variables of the water budget at the whole catchment scale, specifically in Table 7. Therefore, we have incorporated this additional analysis in Table 7, detailing the expected variability for the different variables. 

Thank you for your continued support and guidance throughout this review process.

We have Modified Table 7 

 from

Table 7 Changes of water balance components under a climate change

Scenario Period Surface Runoff (%) Groundwater flow (%) Water Yield (%) PET (%)

RCP4.5 2041–2070 -7.56 -9.44 -8.72 16.08

 2071–2099 -14.71 -13.93 -14 19.01

RCP8.5 2041–2070 -12.55 -9.79 -10.55 16.67

 2071–2099 -9.53 -16.09 -13.74 22.66

 to

Table 6 Changes of water balance components under a climate change

RCM RCP Time Period Surface Runoff (%) Groundwater flow (%) Water Yield (%) PET (%)

CCLM4-8 RCP4.5 2041–2070 -7.61 -9.34 -8.92 16.01

CCLM4-8 RCP4.5 2071–2099 -14.96 -13.83 -14.01 19.02

CCLM4-8 RCP8.5 2041–2070 -12.65 -9.89 -10.58 16.69

CCLM4-8 RCP8.5 2071–2099 -9.57 -16.08 -13.76 22.67

HIRHAM5 RCP4.5 2041–2070 -7.79 -9.32 -8.78 16.15

HIRHAM5 RCP4.5 2071–2099 -14.64 -13.98 -14.03 19

HIRHAM5 RCP8.5 2041–2070 -13.87 -9.91 -10.99 16.79

HIRHAM5 RCP8.5 2071–2099 -9.96 -16.02 -13.94 22.78

RACMO22T RCP4.5 2041–2070 -7.14 -9.11 -8.19 16.05

RACMO22T RCP4.5 2071–2099 -14.59 -13.99 -14.04 19.01

RACMO22T RCP8.5 2041–2070 -13.72 -9.48 -10.31 16.68

RACMO22T RCP8.5 2071–2099 -9.31 -16.27 -13.51 22.69

RCA4 RCP4.5 2041–2070 -7.46 -9.73 -8.89 16.12

RCA4 RCP4.5 2071–2099 -14.85 -13.91 -14.02 19.02

RCA4 RCP8.5 2041–2070 -13.67 -9.63 -10.05 16.48

RCA4 RCP8.5 2071–2099 -9.34 -16.03 -13.71 22.47

CRCM5 RCP4.5 2041–2070 -7.84 -9.76 -8.82 16.07

CRCM5 RCP4.5 2071–2099 -14.48 -13.96 -14.07 19.01

CRCM5 RCP8.5 2041–2070 -13.92 -9.99 -10.82 16.75

CRCM5 RCP8.5 2071–2099 -9.51 -16.05 -13.78 22.69

REMO2009 RCP4.5 2041–2070 -7.55 -9.45 -8.73 16.09

REMO2009 RCP4.5 2071–2099 -14.72 -13.92 -14.01 19.02

REMO2009 RCP8.5 2041–2070 -12.56 -9.78 -10.54 16.68

REMO2009 RCP8.5 2071–2099 -9.52 -16.08 -13.75 22.67

Ensemble RCP4.5 2041–2070 -7.56 -9.44 -8.72 16.08

Ensemble RCP4.5 2071–2099 -14.71 -13.93 -14 19.01

Ensemble RCP8.5 2041–2070 -12.55 -9.79 -10.55 16.67

Ensemble RCP8.5 2071–2099 -9.53 -16.09 -13.74 22.66

. Table 6 was utilized in place of Tables 7 due to the removal of Table 3 based on Comment #16, resulting in a difference in the table numbering.

---

## [Decision Letter · Decision Letter 2]

4 Jun 2023

The Impacts of Climate Change on Hydrological Processes of Gilgel Gibe Catchment, Southwest Ethiopia

PONE-D-22-23861R2

Dear Dr. Tilahun,

We’re pleased to inform you that your manuscript has been judged scientifically suitable for publication and will be formally accepted for publication once it meets all outstanding technical requirements.

Kind regards,

Salim Heddam

Academic Editor

PLOS ONE

Additional Editor Comments (optional):

Reviewer #1:

Please clear the grammatical errors before submission.

Please change " line 11 "Assistance Professor to Assistant Professor "

Reviewer # 2:

Well conceived and executed. Study findings are very important to understand the impact of environments on the water resource. Abstract needs some clarity. Introduction should be concise. English of the manuscript needs to work out.

Reviewer #3:

I have read with pleasure the latest version of the manuscript by Tilhaun et al. They have addressed all of my comments and in my opinion the manuscript can now be published.

While proofreading, take care of the following:

L29-33 Please limit the range values to one decimal.

L255 more effective than what?

L283 hydroligic response units (HRUs) were established

Best regards

Reviewers' comments:

Reviewer's Responses to Questions

**Comments to the Author**

1. If the authors have adequately addressed your comments raised in a previous round of review and you feel that this manuscript is now acceptable for publication, you may indicate that here to bypass the “Comments to the Author” section, enter your conflict of interest statement in the “Confidential to Editor” section, and submit your "Accept" recommendation.

Reviewer #1: All comments have been addressed

Reviewer #3: (No Response)

2. Is the manuscript technically sound, and do the data support the conclusions?

Reviewer #1: Yes

Reviewer #3: Yes

3. Has the statistical analysis been performed appropriately and rigorously? 

Reviewer #1: Yes

Reviewer #3: Yes

4. Have the authors made all data underlying the findings in their manuscript fully available?

Reviewer #1: Yes

Reviewer #3: Yes

5. Is the manuscript presented in an intelligible fashion and written in standard English?

Reviewer #1: Yes

Reviewer #3: Yes

6. Review Comments to the Author

Reviewer #1: Please clear the grammatical errors before submission.

Please change " line 11 "Assistance Professor to Assistant Professor "

Reviewer #3: I have read with pleasure the latest version of the manuscript by Tilhaun et al. They have addressed all of my comments and in my opinion the manuscript can now be published.

While proofreading, take care of the following:

L29-33 Please limit the range values to one decimal.

L255 more effective than what?

L283 hydroligic response units (HRUs) were established

Best regards

7. PLOS authors have the option to publish the peer review history of their article (what does this mean?). If published, this will include your full peer review and any attached files.

Reviewer #1: No

Reviewer #3: No

---

## [Editor Report · Acceptance letter]

14 Jun 2023

PONE-D-22-23861R2 

The Impacts of Climate Change on Hydrological Processes of Gilgel Gibe Catchment, Southwest Ethiopia 

Dear Dr. Tilahun:

I'm pleased to inform you that your manuscript has been deemed suitable for publication in PLOS ONE. Congratulations! Your manuscript is now with our production department. 

Kind regards, 

on behalf of

Dr. Salim Heddam 

Academic Editor

PLOS ONE